# DAD-SFT: Dual Attention Distillation for Lightweight UAV Vision-Language Navigation

## Abstract

In recent years, Unmanned Aerial Vehicle (UAV) Vision-Language Navigation (VLN) has attracted increasing attention due to its broad applications in scenarios such as autonomous inspection and emergency rescue. Large-scale Vision-Language Models (VLMs) demonstrate strong cross-modal understanding and reasoning capabilities; however, their massive parameter size and computational demands hinder their deployment on resource-constrained devices. Although lightweight models facilitate efficient deployment, their performance and generalization ability remain limited. To address this challenge, we propose a **Dual Attention Distillation into Supervised Fine-Tuning (DAD-SFT)** framework. First, Cross-Modal Attention Distillation (CAD) is employed to guide the student model in aligning its semantic focus patterns with those of a powerful teacher model, thereby enhancing its cross-modal perception ability. Meanwhile, we introduce a Contrastive Attention Alignment (CAA) that constructs diverse types of negative samples to strengthen the model's discriminative capability, which in turn improves generalization under complex scenarios. Systematic evaluations on the CityNav benchmark demonstrate that our method consistently outperforms mainstream baselines in terms of navigation accuracy, cross-scene generalization, and deployment efficiency, showcasing strong overall performance and practical potential. Our code is publicly available for reproducibility.[1]

## 1 Introduction

Unmanned Aerial Vehicle (UAV) Vision-Language Navigation (VLN) has emerged as a key research focus, driven by its potential in diverse real-world applications including infrastructure inspection, disaster response, and ecological surveillance. As a multimodal fusion task, UAV VLN requires an agent to integrate visual perception and natural language instructions for path planning and decision-making in dynamic environments. Vision-Language Models (VLMs), with their strong cross-modal alignment and reasoning capabilities, have demonstrated remarkable performance in this task, enabling UAVs to navigate more accurately and flexibly.

However, existing large-scale VLMs typically contain an enormous number of parameters, resulting in substantial computational and storage overhead, which severely hinders their deployment on resource-constrained devices. In contrast, lightweight models offer advantages in terms of deployment cost and inference latency, making them more suitable for real-world UAV applications. Nevertheless, due to their limited model capacity, lightweight models often struggle with cross-modal alignment, semantic understanding, and reasoning, which leads to high navigation failure rates and poor cross-scene generalization. These shortcomings make it difficult to meet the demands of high accuracy and robustness. Therefore, how to effectively transfer the perceptual and reasoning capabilities of large models into lightweight architectures, while maintaining deployment efficiency, has become a key challenge in UAV VLN research.

To address the above issues, we propose a novel framework named **Dual Attention Distillation into Supervised Fine-Tuning (DAD-SFT)**. This method leverages **Cross-Modal Attention Distillation (CAD)** to guide the student model in learning the teacher model's semantic focus patterns, and incorporates **Contrastive Attention Alignment (CAA)** to enhance the discriminative ability of the model

---

[1] https://anonymous.4open.science/r/DAD-SFT

using positive and negative samples. Through the synergy of perceptual transfer and discriminative optimization, DAD-SFT significantly improves the cross-modal understanding and generalization ability of lightweight models, enabling efficient and robust navigation on resource-limited devices. To summarize, our work offers the following four key contributions.

**First**, we propose a unified framework, DAD-SFT, which integrates knowledge distillation and contrastive learning to strike a balance between performance and efficiency for lightweight models.

**Second**, we design a fine-grained distillation approach based on cross-modal attention to precisely transfer the perceptual and alignment capabilities of the teacher model to the student model.

**Third**, we introduce a CAA strategy that constructs diverse negative samples and incorporates Contrastive Learning (CL) to enhance the student model's discriminative power and cross-scene generalization ability.

**Finally**, we conduct comprehensive evaluations on the CityNav dataset, demonstrating that our method outperforms various baselines in terms of performance, even surpassing the teacher model in some cases, while also exhibiting superior efficiency in terms of memory consumption and inference latency, validating its advantages in both effectiveness and deployability.

## 2 RELATED WORK

In recent years, the VLN task has been gradually extended to UAV scenarios, supporting practical applications such as autonomous inspection and emergency response Wu et al. (2024); Wang et al. (2024). Although VLMs have demonstrated outstanding cross-modal understanding and reasoning capabilities in this task, their large model size and high inference cost hinder deployment on edge devices Feng et al. (2025); Vasu et al. (2025); Qiao et al. (2025). To balance performance and efficiency, researchers have proposed lightweight solutions including architecture re-design, sparsification, and knowledge distillation, and have also explored CL mechanisms to enhance model generalization Ye et al. (2025); Zhang et al. (2024a); Jang et al. (2025); Ge et al. (2025). However, these methods still lack systematic validation in complex UAV VLN settings. Building upon this foundation, we propose the DAD-SFT framework, which improves the performance of lightweight models while maintaining deployment feasibility. A more detailed discussion of related work can be found in Appendix B.

## 3 METHODOLOGY

### 3.1 PROBLEM FORMULATION

This study focuses on the task of UAV VLN, which aims to enable a UAV to navigate within a three-dimensional environment by following natural language instructions. The task requires the agent to perceive, understand, and make decisions based on multimodal inputs in order to complete the navigation from the starting point to the specified target location. Formally, each instance in the UAV VLN task can be represented as a triplet $(I, D, E)$:

- $I$: the initial state of the UAV, including its starting position and orientation;
- $D$: a natural language instruction, typically describing the target location and its surrounding semantic landmarks;
- $E$: the environment, which includes real-world spatial structures and rich semantic elements, such as roads, buildings, and objects (e.g., vehicles).

During navigation, the UAV receives first-person visual observations from the environment, including RGB images and depth information. It also has access to a 2D map that is aligned with actual geographic information. Based on these inputs, the agent is expected to execute a sequence of discrete actions $A = \{a_1, a_2, ..., a_T\}$. The action space comprises a set of basic operations, including **move forward**, **turn left**, **turn right**, **ascend**, **descend**, and **stop**. The UAV may choose to perform the **stop** action when it determines that it is sufficiently close to the target region. A navigation episode is considered successful if the final stopping position falls within a predefined threshold distance (e.g., 20 meters) from the target location.

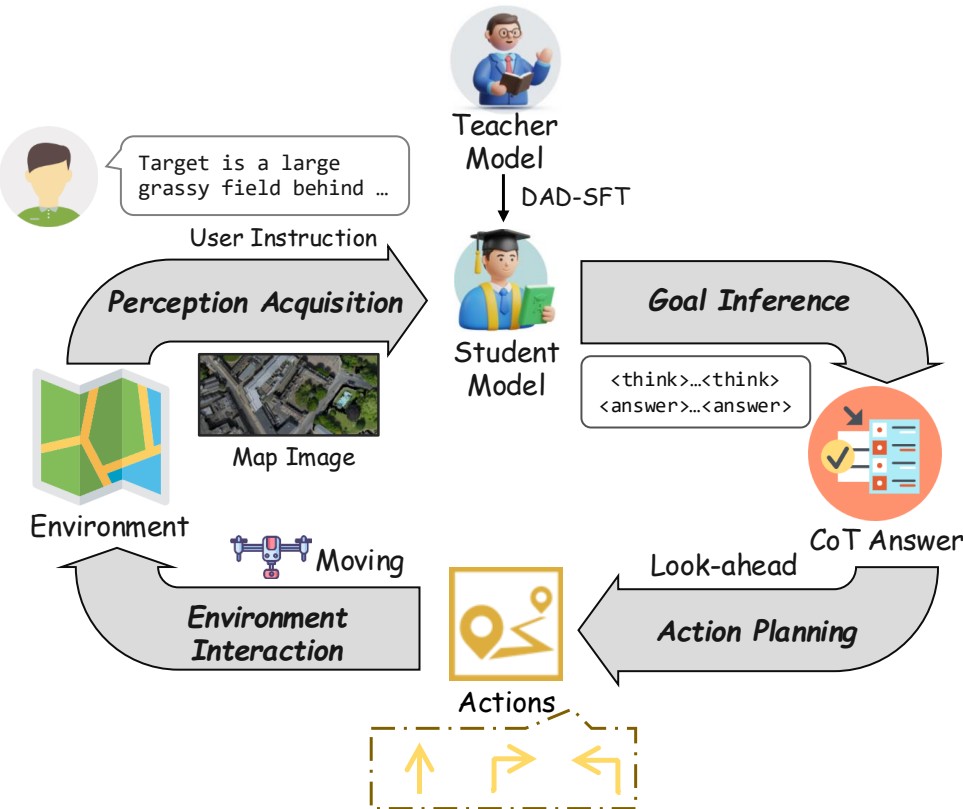

Figure 1: The closed-loop **UAV VLN system pipeline**, comprising four modules: **Perception Acquisition**, which processes multi-modal inputs including visual and textual data; **Goal Reasoning**, where the student model infers the target location under the guidance of a teacher model via Dual Attention Distillation into Supervised Fine-Tuning (DAD-SFT); **Action Planning**, which generates action sequences using a Look-ahead mechanism; and **Environment Interaction**, where the UAV executes actions and updates its state within the environment.

## 3.2 OVERALL SYSTEM PIPELINE FOR UAV VLN

To address the deployment challenges in resource-constrained UAV VLN scenarios, we design a closed-loop navigation pipeline inspired by previous works (Anderson et al., 2018; Liu et al., 2023; Lee et al., 2024). As illustrated in Figure 1, the proposed system pipeline consists of four key modules: **Perception Acquisition**, **Goal Inference**, **Action Planning**, and **Environment Interaction**.

**1. Perception Acquisition**: The system receives multi-modal inputs, including RGB images from the UAV's first-person view, semantic maps, and natural language instructions.

**2. Goal Inference**: This module combines visual perception with natural language instructions to identify the key semantic regions and target location. To improve the lightweight model's cross-modal understanding and generalization capabilities, we introduce the **DAD-SFT**, which transfers knowledge from a high-capacity teacher model to a compact student model.

**3. Action Planning**: Based on the inferred goal, the Look-ahead mechanism (Liu et al., 2023) is employed to predict the optimal sequence of actions, ensuring both efficiency and rationality in trajectory planning.

**4. Environment Interaction**: The UAV executes the actions and updates its internal state, while the environmental information is simultaneously refreshed to reflect the new context.

This process iterates continuously until a **stop** action is triggered or a predefined maximum number of steps is reached.

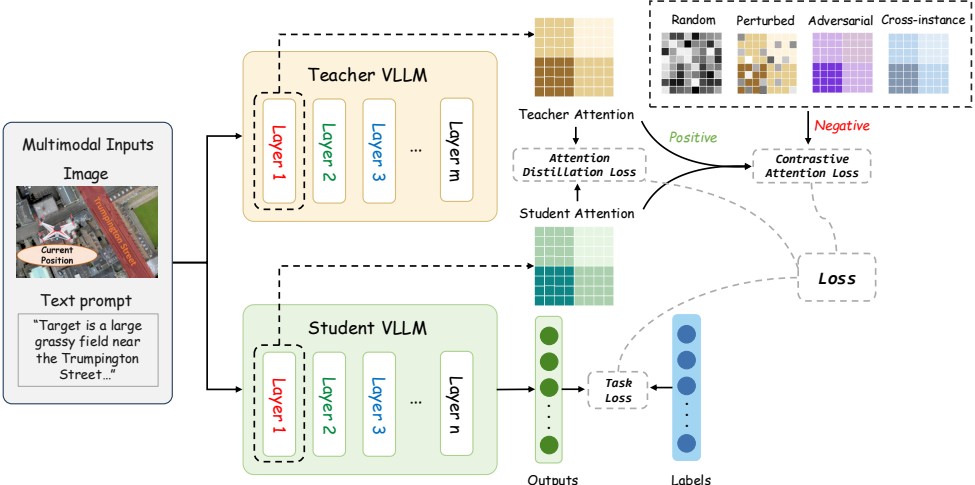

Figure 2: **The DAD-SFT framework.** Multi-modal inputs are fed into both the teacher and student VLMs. The student model aligns its cross-modal attention with the teacher via **Attention Distillation Loss**, while **Contrastive Attention Loss** leverages teacher attention as a positive sample and four types of negative samples. The overall training objective combines **Task Loss** with auxiliary objectives to improve both performance and generalization.

### 3.3 DUAL ATTENTION DISTILLATION INTO SUPERVISED FINE-TUNING

In the UAV VLN task, **Goal Inference** is a pivotal step toward achieving high-precision navigation. This phase requires the model to jointly reason over current multimodal perceptual inputs to accurately localize the target region, based on which effective path planning strategies can be formulated. To enhance the performance of lightweight models at this stage, we perform Supervised Fine-Tuning (SFT) on the Goal Inference module of the system pipeline (as described in Section 3.2).

However, naive SFT approaches rely solely on task-level labels for optimization, which are insufficient to compensate for the limitations of lightweight models in perception accuracy and generalization ability. To address this challenge, we propose a novel framework termed **DAD-SFT**. As illustrated in Figure 2, both visual and textual inputs are simultaneously fed into the teacher and student models. During training, the cross-modal attention output from the teacher model serves as a supervisory signal for distillation, guiding the student model to attend to critical semantic regions. To further reinforce the model's learning, we introduce CAA. The teacher's attention is used as a positive sample, while a diverse set of negative attention distributions is constructed to serve as contrasts. This contrastive supervision enhances the student model's ability to distinguish between "what should be attended to" and "what should not," promoting robust semantic focus under challenging scenarios. By jointly optimizing the Task Loss, Attention Distillation Loss, and Contrastive Attention Loss, the DAD-SFT framework substantially enhances the overall performance of the student model, striking a balance between accuracy, generalization, and deployment efficiency.

#### 3.3.1 INPUTS, PROMPT DESIGN, AND OUTPUTS

(1) Model Inputs

The inputs to **DAD-SFT** consist of the following two components:

- **Semantic Map**: A semantic map that integrates multimodal information. It contains annotations indicating the UAV's current position and orientation, the first-person field of view, and the locations of key landmarks.
- **Textual Information**: Text descriptions that provide the UAV's current state, including its position and orientation, as well as the target description expressed in natural language.

(2) Prompt Template

To enhance the model's reasoning ability and the interpretability of its decisions, we follow the design of FlightGPT (Cai et al., 2025b) and adopt a Chain-of-Thought (CoT) prompt template. The complete prompt template is provided in Appendix C.

(3) Model Outputs

Conforming to the above prompt specification, the model produces:

- **Intermediate Reasoning.** Within the `<think>` tags, the model outputs a stepwise rationale that usually includes semantic parsing of the instruction, identification of relevant landmark(s), and spatial reasoning based on visual and semantic cues.
- **Final Prediction.** Within the `<answer>` tags, the model outputs the predicted target coordinates $(x, y)$.

### 3.3.2 CONSTRUCTION OF CoT DATA

Due to the lack of high-quality annotated reasoning data tailored for the UAV VLN task in existing public datasets, we adopt the high-performance VLM **Qwen2.5-VL-32B** as a data generator. By integrating the input format and prompt templates designed in Section 3.3.1, we generate training samples that incorporate CoT reasoning. These samples provide intermediate reasoning supervision signals, which are crucial for guiding model optimization beyond final prediction accuracy. To ensure data quality, we apply a series of strict filtering and enhancement strategies: **(1) Format Validation**: Samples that do not satisfy the prompt required or exhibit abnormal output behaviors are excluded. **(2) Precision Filtering**: Samples are removed if the predicted target location deviates from the ground truth by more than 20 meters. **(3) Answer Replacement**: For retained samples, the model-predicted coordinates are replaced with the ground-truth coordinates.

After applying the above procedures, we obtain a total of 2,300 high-quality training samples. These samples span diverse navigation scenarios and semantic goals. Statistical analysis of the CoT dataset is provided in Appendix J.

### 3.3.3 DUAL ATTENTION DISTILLATION MECHANISM

We refer to our mechanism as **Dual Attention Distillation**, which consists of two complementary components: **Cross-Modal Attention Distillation (CAD)** and **Contrastive Attention Alignment (CAA)**. While both are attention-focused, the former provides soft supervision from a teacher model, and the latter enhances discriminability via contrastive learning with negative samples.

**(1) Cross-Modal Attention Distillation**
In UAV VLN tasks, cross-modal attention serves as a critical indicator of where the model "attends" during the execution of navigation instructions, reflecting the model's capability for vision-language alignment. Large-scale VLMs typically possess stronger cross-modal perception and reasoning abilities. Their attention distributions often precisely focus on semantically critical regions. Therefore, transferring such attention patterns to lightweight student models is expected to significantly enhance their cross-modal perception capability.

To this end, we propose the **CAD**. This method aims to provide the student model with fine-grained supervision signals via the cross-modal attention distribution. Specifically, we extract the text-to-vision cross-modal attention from the teacher model and use it as the distillation target to guide the student model in learning a similar semantic attention pattern.

Prior studies (Feng et al., 2025; Zhou et al., 2025; Chen et al., 2025) have shown that the cross-attention in shallow layers of VLMs can establish stable semantic correlations between visual patches and language tokens. These shallow layers often outperform deeper layers in terms of cross-modal alignment, which is particularly beneficial for fundamental vision tasks such as object grounding. In contrast, the cross-modal attention in deeper layers tends to focus on more complex semantic integration and task-specific reasoning, and when used directly for distillation, it may cause the student model to overfit specific task behaviors (Elnoor et al., 2025; Li et al., 2024). Hence, in our implementation, we align only cross-modal attention distributions of the first layer, encouraging the student model to learn stable and transferable semantic focus patterns from the teacher model. This decision is not only motivated by prior works, but also supported by our own empirical study.

We conduct an ablation experiment comparing distillation at different layers (first vs. last), and observe that using the first layer attention yields significantly better performance. Detailed results are provided in Appendix D.

To quantitatively evaluate the alignment of attention distributions between the student and teacher, we adopt the Kullback-Leibler (KL) divergence as the distance metric between distributions. The attention distillation loss is defined as follows:

$$\mathcal{L}_{\text{attn}} = \sum_{l=1}^{L} w_l \cdot D_{\text{KL}} \left( A_{\text{teacher}}^{(l)} \parallel A_{\text{student}}^{(l)} \right),$$ (1)

where $A^{(l)}$ denotes the cross-modal attention distribution at the $l$-th layer, and $w_l$ is a layer-specific loss weight hyperparameter. In our work, we align only the first layer's cross-modal attention using a KL divergence loss.

### (2) Contrastive Attention Alignment

Although attention knowledge distillation has demonstrated promising performance in enhancing cross-modal perception for lightweight models, existing methods primarily emphasize regions the student model **should attend to**, while neglecting regions that **should not be attended to** or are **incorrectly focused**. Such supervision lacks negative signals and tends to result in ambiguous attention and limited discrimination capability, especially when confronted with complex instructions or multi-object scenarios, thereby restricting the model's generalization ability.

To address this limitation, we introduce a CL mechanism and propose the **CAA**, aiming to explicitly enhance the discriminability of attention distributions via positive and negative sample construction. Specifically, we treat the cross-modal attention extracted from the teacher model as *positive samples*, and construct four representative types of *negative samples*:

**1. Random Attention**: Attention maps generated via random initialization, simulating the scenario of no semantic focus.

**2. Perturbed Attention**: Slightly noisy variants of the teacher's original attention maps, creating "similar but incorrect" pseudo-attention distributions.

**3. Adversarial Attention**: Pseudo-instructions are constructed by semantically flipping key attributes in the original instructions (e.g., changing "left of the red roof" to "right of the red roof"). The corresponding attention distributions are generated using the teacher model. These adversarial instructions are created using LLM (GPT-4o), with prompt templates provided in Appendix E.

**4. Cross-instance Attention**: Teacher attention maps randomly sampled from other instances in the training set.

We adopt a contrastive loss in the form of **InfoNCE** to optimize the student model by maximizing the similarity between the student attention and the positive sample, while minimizing the similarity with all negative samples. The contrastive loss is defined as:

$$\mathcal{L}_{\text{contrast}} = -\log \frac{\exp\left(\text{sim}(A_{\text{stu}}, A_{\text{pos}})/\tau\right)}{\exp\left(\text{sim}(A_{\text{stu}}, A_{\text{pos}})/\tau\right) + \sum_k \exp\left(\text{sim}(A_{\text{stu}}, A_{\text{neg}}^{(k)})/\tau\right)}.$$ (2)

Here, $A_{\text{stu}}$ denotes the attention map from the student model, $A_{\text{pos}}$ is the positive attention sample from the teacher, and $A_{\text{neg}}^{(k)}$ refers to the $k$-th negative sample. The function $\text{sim}(\cdot)$ represents a similarity metric, implemented as *cosine similarity*, and $\tau$ is the temperature parameter that controls the sensitivity of the distribution.

This contrastive attention loss introduces supervision signals from negative examples, allowing the student model to more clearly distinguish between "attended" and "unattended" regions. As a result, it builds more discriminative attention representations and enhances the student model's adaptability and generalization capability in diverse environments.

### 3.3.4 TRAINING OBJECTIVE AND OPTIMIZATION

Building upon the previously introduced Dual Attention Distillation Mechanism (Section 3.3.3), we integrate three complementary types of supervision signals—Task Execution, CAD, and CAA—into a unified SFT paradigm. This leads to the formulation of the **DAD-SFT** framework. Within this framework, the overall objective function is composed of the following three loss terms:

- **Task Loss** $\mathcal{L}_{\text{task}}$: This term drives the model to accomplish the core navigation task. Based on the task setting, it employs the commonly used *Next Token Prediction* (NTP) loss from language modeling, enabling the model to learn path planning and goal localization capabilities.

- **Attention Distillation Loss** $\mathcal{L}_{\text{attn}}$: This loss utilizes the cross-modal attention distribution from the first layer of the teacher model as supervision, encouraging the student model to learn a consistent attention pattern. The discrepancy between student and teacher attention distributions is measured via KL divergence. The objective is to transfer the teacher's cross-modal perceptual capability to the student, thereby enhancing the cross-modal alignment of the lightweight model.

- **Contrastive Attention Loss** $\mathcal{L}_{\text{contrast}}$: By constructing positive and negative samples, this term introduces an InfoNCE-based CAA objective. It maximizes similarity with positives while minimizing similarity with negatives, guiding the student model to more clearly distinguish between semantically relevant ("should attend to") and irrelevant ("should not attend to") regions. This enhances the discriminability and generalization of the model's attention representations.

The final training objective is defined as a weighted combination of the three loss functions:

$$\mathcal{L} = \mathcal{L}_{\text{task}} + \lambda_{\text{attn}}\mathcal{L}_{\text{attn}} + \lambda_{\text{contrast}}\mathcal{L}_{\text{contrast}},$$

where $\lambda_{\text{attn}}$ and $\lambda_{\text{contrast}}$ are hyperparameters that control the relative contribution of each supervision signal.

By jointly optimizing the above losses, the student model not only acquires cross-modal perceptual abilities from the teacher model but also enhances its capacity to distinguish distracting regions at the attention level in complex scenarios.

## 4 EXPERIMENTS

### 4.1 DATASETS

We conduct our experiments on the CityNav benchmark dataset (Lee et al., 2024). CityNav offers a city-scale simulation environment that includes rich semantic elements such as roads, buildings, and landmarks, enabling UAVs to perform VLN tasks in realistic and complex urban scenes. The navigation instructions are presented in natural language, typically describing the target location along with surrounding landmarks, which brings the task closer to real-world applications. The dataset provides standard splits including **Validation Seen**, **Validation Unseen**, and **Test Unseen**, allowing for comprehensive evaluation of the model's capability in both In-Distribution (ID) and Out-Of-Distribution (OOD) environments. Following CityNav's official evaluation protocol, we report four metrics: Navigation Error (NE), Success Rate (SR), Oracle Success Rate (OSR), and Success weighted by Path Length (SPL). Detailed definitions of these metrics are provided in Appendix F.

### 4.2 BASELINES

We compare our proposed approach against two representative categories of baseline models. The first category includes conventional methods, such as Random, sequence-to-sequence (Seq2Seq) (Anderson et al., 2018), Cross-Modal Attention (CMA) (Liu et al., 2023), and Map-based Goal Prediction (MGP) (Lee et al., 2024), which are widely adopted in the UAV VLN task and serve as reliable references for performance comparison. The second category comprises strong-performing VLMs, including LLaMA-3.2-11B-Vision (Grattafiori et al., 2024), Qwen2.5-VL (Bai et al., 2025), and GPT-4o (OpenAI et al., 2024), which exhibit remarkable capabilities in cross-modal understanding and reasoning, but also incur significantly higher computational overhead and deployment costs. Additionally, to evaluate the effectiveness of Reinforcement Learning (RL) in lightweight models, we introduce **Naive RL** as a supplementary baseline. The method is trained

entirely within the RL framework, where the reward combines *goal prediction accuracy* and *output format compliance*. The introduction to these baseline models is provided in Appendix G. All baseline models are trained and evaluated under the same input configuration, including the use of semantic maps, to ensure fair comparison.

### 4.3 TRAINING DETAILS

The training process is conducted based on the Qwen2.5-VL. The teacher model is Qwen2.5-VL-32B, which possesses strong cross-modal perception and reasoning capabilities. The student model is the lightweight Qwen2.5-VL-3B, chosen for higher inference efficiency. Both models share the same architecture, facilitating alignment during CAD and CAA. All experiments are performed on 8*A100 GPUs. The specific hyperparameter configurations, along with detailed justifications for their selection, are provided in Appendix H.

## 5 RESULTS AND ANALYSIS

### 5.1 MAIN RESULTS

We compare a range of baselines, and the experimental results are shown in Table 1. The key findings are summarized as follows.

**First**, our method consistently outperforms traditional approaches (e.g., Seq2Seq, CMA, MGP) and zero-shot LLMs (e.g., GPT-4o, LLaMA-3.2-11B-Vision) across all evaluation metrics, and achieves performance that is comparable to or even surpasses the teacher model (Qwen2.5-VL-32B) in several cases. For instance, in terms of SR, our model achieves **12.73%**, **10.43%**, and **12.96%** on the Validation Seen, Validation Unseen, and Test Unseen splits, respectively, which not only exceeds MGP's performance (8.69% / 5.84% / 6.38%) but also surpasses the teacher model's results (12.65% / 10.12% / 11.98%), demonstrating superior accuracy and robustness.

**Second**, under the lightweight model setting (3B), the model trained with our DAD-SFT framework significantly outperforms the original student model as well as models trained with Naive SFT, Naive RL, CAD-only, and CAA-only strategies across all evaluation metrics. This validates the synergistic effect of jointly applying both CAD and CAA.

**Third**, our method maintains a clear advantage in OOD scenarios. On the Test Unseen split, our model achieves the best performance across all four metrics: NE, SR, OSR, and SPL, indicating strong generalization ability and robustness in unseen environments.

Table 1: Quantitative comparison of baseline models and proposed methods on the CityNav dataset across three evaluation splits. **Bold** numbers denote the best results, and underlined numbers denote the second-best.

| Method | Validation Seen | | | | Validation Unseen | | | | Test Unseen | | | |
| --- | --- | --- | --- | --- | --- | --- | --- | --- | --- | --- | --- | --- |
| | NE↓ | SR↑ | OSR↑ | SPL↑ | NE↓ | SR↑ | OSR↑ | SPL↑ | NE↓ | SR↑ | OSR↓ | SPL↑ |
| Random | 222.30 | 0.00 | 1.15 | 0.00 | 223.00 | 0.00 | 0.90 | 0.00 | 208.80 | 0.00 | 1.44 | 0.00 |
| Seq2Seq+GSM | 58.5 | 8.43 | 17.31 | 7.28 | 78.6 | 5.13 | 10.90 | 4.65 | 98.1 | 3.81 | 13.92 | 2.79 |
| CMA+GSM | 68.0 | 6.25 | 13.28 | 5.40 | 75.9 | 4.38 | 9.29 | 3.90 | 94.6 | 4.68 | 12.01 | 4.05 |
| MGP | 59.70 | 8.69 | **35.51** | 8.28 | 75.10 | 5.84 | **22.19** | 5.56 | 93.80 | 6.38 | 26.04 | 6.08 |
| LLaMA-3.2-11B-Vision | 198.90 | 1.16 | 5.16 | 1.06 | 215.10 | 0.50 | 4.35 | 0.46 | 191.10 | 1.26 | 4.59 | 1.15 |
| GPT-4o | 155.80 | 2.42 | 9.62 | 2.17 | 170.40 | 2.17 | 7.77 | 1.98 | 144.40 | 3.90 | 11.79 | 3.42 |
| Qwen2.5-VL-7B | 116.10 | 4.72 | 12.89 | 4.15 | 123.20 | 5.52 | 13.98 | 4.92 | 124.60 | 4.59 | 12.75 | 3.99 |
| Qwen2.5-VL-32B (teacher) | 84.70 | 12.65 | 24.14 | 11.30 | 91.90 | 10.12 | 20.52 | 9.00 | 83.28 | 11.98 | 23.48 | 10.76 |
| Qwen2.5-VL-3B (student) | 171.16 | 1.48 | 3.36 | 1.44 | 181.64 | 1.17 | 3.36 | 1.10 | 165.56 | 1.45 | 2.90 | 1.41 |
| Qwen2.5-VL-3B (Naive SFT) | 121.99 | 4.60 | 13.65 | 4.18 | 120.89 | 5.57 | 14.22 | 5.02 | 123.63 | 5.79 | 16.49 | 5.34 |
| Qwen2.5-VL-3B (Naive RL) | 98.83 | 9.16 | 19.18 | 8.06 | 105.02 | 6.51 | 17.76 | 5.67 | 104.53 | 10.00 | 21.83 | 8.98 |
| Qwen2.5-VL-3B (CAD-only) | 97.84 | 9.33 | 20.25 | 9.26 | 101.94 | 8.09 | 19.95 | 7.18 | 115.86 | 8.03 | 18.07 | 7.24 |
| Qwen2.5-VL-3B (CAA-only) | 106.82 | 8.86 | 19.81 | 7.44 | 99.76 | 9.45 | 20.58 | 7.67 | 108.43 | 9.79 | 20.07 | 8.25 |
| Qwen2.5-VL-3B (DAD-SFT) | 71.51 | **12.73** | 25.17 | **11.96** | 70.09 | **10.43** | 22.79 | **9.61** | 76.94 | **12.96** | **26.78** | **12.02** |

### 5.2 ABLATION STUDY

To evaluate the effectiveness of each key component in the proposed DAD-SFT framework, we conduct an ablation study with four configurations: **(1) Naive SFT**, trained only with the Task Loss; **(2) CAD-only**, which builds upon Naive SFT by incorporating Attention Distillation Loss;

**(3) CAA-only**, which builds upon Naive SFT by incorporating Contrastive Attention Loss; and **(4) DAD-SFT**, which combines Task Loss, Attention Distillation Loss, and Contrastive Attention Loss. The results, as shown in Table 1, lead to the following three key observations:

**CAD improves imitation ability and enhances multi-modal perception.** Integrating CAD significantly strengthens the student model's cross-modal perception. Compared to Naive SFT, CAD notably reduces the NE in seen environments (e.g., Validation Seen) from 121.99 to 97.84, while increasing the SR from 4.60% to 9.33%. These results indicate that the student model better inherits the semantic focus patterns of the teacher model.

**CAA enhances discriminative ability and improves generalization.** CAA leverages positive and negative samples to guide the student model in distinguishing between relevant and irrelevant regions. On the challenging Test Unseen split, SR improves from 5.79% to 9.79% and SPL increases from 5.34 to 8.25 compared to Naive SFT, demonstrating that the model gains better generalization and robustness in complex scenarios.

**CAD and CAA are complementary, and DAD-SFT achieves the best overall performance.** When both mechanisms are combined, DAD-SFT consistently outperforms all ablation variants across all data splits and evaluation metrics. This highlights the synergistic effect between perceptual alignment and discriminative supervision, and verifies the effectiveness of the proposed design.

We also conduct two additional ablation studies to further validate the robustness and design of our framework: (1) a grid search sensitivity analysis on the loss weights $\lambda_{attn}$ and $\lambda_{contrast}$; and (2) a component-wise study on different negative sample types used in CAA. The detailed results and discussion are provided in Appendix K.

## 5.3 EFFICIENCY

While maintaining high performance, the ability to deploy models efficiently on resource-constrained devices is key for the practical application of UAV VLN. To evaluate the deployment potential of our proposed method, we perform a systematic comparison of multiple representative models across three dimensions—hardware requirements, memory usage, and inference latency. The detailed results are provided in Appendix I. The following key observations can be made:

**Large-scale models are difficult to deploy.** The teacher model Qwen2.5-VL-32B achieves high accuracy, but requires more than 70GB of GPU memory and has an inference latency of 53.42 s/step. It can only run on high-end A100 GPUs, making it impractical for edge deployment.

**Cloud-based models are less controllable.** Although GPT-4o avoids local memory usage by relying on cloud inference, it still exhibits a latency of 9.73 s/step. The absence of strong guarantees for privacy and real-time responsiveness limits its practicality in UAV applications.

**Traditional models are lightweight but underperform.** Models like Seq2Seq, CMA, and MGP require less than 1GB of memory and have low latency (1.00/1.01/0.82 s/step), but their performance falls significantly behind VLMs, making them unsuitable for complex environments.

**Our method balances performance and efficiency.** The proposed lightweight model runs on a single RTX 4090 with only 13.53GB GPU memory usage and 6.82 s/step inference latency. It significantly outperforms mainstream VLMs in efficiency, demonstrating excellent deployment potential and practical value.

## 6 CONCLUSION

In this work, we address the limitations of lightweight models in UAV VLN, particularly their weak cross-modal perception ability and poor generalization. We propose the DAD-SFT framework, which combines CAD with CAA. This approach guides the student model to learn the teacher model's semantic focusing patterns and leverages diverse negative samples with CL to enhance its discriminative capability. Experimental results demonstrate that our method outperforms various mainstream baselines in terms of accuracy, generalization, and deployment efficiency. This study provides a new perspective for exploring the lightweighting and deployability of VLMs, while also offering preliminary validation and practical insights for improving their generalization in dynamic and complex environments.

# 7 ETHICS STATEMENT

**Potential dual-use concerns.** Although our work is intended for socially beneficial applications such as infrastructure inspection and disaster response, the proposed UAV VLN system could potentially be misused for surveillance, target tracking, or other purposes that violate individual privacy or civil liberties. The enhanced semantic grounding between language and vision achieved through attention distillation may unintentionally increase the precision of such misuse. To prevent unethical applications, any real-world deployment should be governed by strict usage boundaries, regulatory frameworks, and human oversight mechanisms.

**Real-world safety risks.** Deploying the proposed system on physical UAV platforms introduces potential risks to public safety. Malfunctions in perception, misinterpretation of natural language instructions, or failure to handle dynamic obstacles may lead to unintended navigation behaviors. Without adequate testing, fail-safe design, and emergency intervention mechanisms, such failures could cause damage to property, disrupt public environments, or even endanger human life. These safety implications must be carefully considered before deploying the system outside controlled settings.

# 8 REPRODUCIBILITY STATEMENT

We have taken several measures to ensure the reproducibility of our work. Detailed descriptions of the system pipeline, model architecture, and optimization strategies are provided in Section 3, including the formulation of our DAD-SFT framework. The construction of training objectives, including the Task Loss, Attention Distillation Loss, and Contrastive Attention Loss, is discussed in Section 3.3.4.

The input modalities and prompt design are presented in Section 3.3.1, while the CoT data generation procedure is described in Section 3.3.2. The prompt templates used for both reasoning and adversarial sample generation are provided in Appendix C and Appendix E, respectively.

Experimental settings, including hardware configuration, evaluation metrics, and all hyperparameters, are summarized in Section 4.3, Appendix F, and Appendix H. Baseline details and efficiency evaluation protocols are documented in Appendix G and I.

To facilitate reproducibility, we provide anonymized source code, model configuration files, and data generation scripts at the following repository: `https://anonymous.4open.science/r/DAD-SFT`. These resources enable full replication of the experiments and all reported results in the paper.

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

## A    THE USE OF LARGE LANGUAGE MODELS

During the preparation of this manuscript, we utilized a large language model (e.g., ChatGPT or similar) solely for purposes of language refinement. Specifically, the model was employed to enhance textual fluency, grammar, and structure. The model generated no content, nor did it participate in experimental design or substantive decision-making. All conceptual contributions and technical analyses were conducted entirely by the authors. The final manuscript has been carefully reviewed and edited by the authors to ensure its originality and correctness.

## B    EXTENDED RELATED WORK

### B.1    UAV-VLN AND LIGHTWEIGHT CHALLENGES

VLN aims to guide an agent to autonomously navigate through environments based on visual perception and natural language instructions Wu et al. (2024). In recent years, this task has been widely introduced into UAV scenarios to support real-world applications such as automated inspection and emergency response Wang et al. (2024). These tasks are typically characterized by high real-time requirements and diverse environments, demanding that the navigation models not only possess strong generalization capabilities but also operate stably on resource-constrained devices.

To meet the requirements of real-world deployment, UAV navigation systems must not only possess multimodal understanding and task execution capabilities, but also maintain computational efficiency to adapt to the constraints of edge computing environments. However, state-of-the-art VLMs have demonstrated exceptional cross-modal understanding and reasoning capabilities in UAV VLN tasks Saxena et al. (2025); Cai et al. (2025b;a), but their large-scale parameter sizes and high inference costs render them impractical for direct deployment on platforms with limited computational resources.

To improve deployability, researchers have proposed a variety of lightweight approaches targeting VLMs, including architectural re-design Ye et al. (2025), pruning and sparsification Zhang et al. (2024a), and knowledge distillation Jang et al. (2025). These methods aim to reduce computational overhead while retaining model performance. Although such strategies have achieved promising results in tasks like image-text retrieval and visual question answering, they still lack systematic evaluation and widespread application in the UAV-VLN setting.

### B.2    KNOWLEDGE DISTILLATION AND CONTRASTIVE LEARNING

To enhance the performance of lightweight models, knowledge distillation has been widely employed to transfer the capabilities of large-scale models to smaller ones Zhang et al. (2024b); Cao et al. (2025). Among these techniques, attention distillation serves as a fine-grained supervision mechanism. By transferring the cross-modal attention distribution from the teacher model, it guides the student model to focus on semantically critical regions, thereby significantly improving its multimodal perception and alignment capabilities Feng et al. (2025); Elnoor et al. (2025). This method has been validated in tasks such as image-text matching Csizmadia et al. (2025) and visual question answering Yang et al. (2025), and has gradually been extended to VLN tasks Elnoor et al. (2025), where it helps enhance the accuracy and stability of navigation path prediction for student models.

Beyond knowledge distillation, some studies have explored the use of CL to further improve the discriminability and generalization ability of models. For instance, ViLTA Wang et al. (2023) introduces a cross-modal contrastive objective during pretraining, encouraging alignment of visual and textual representations within a shared semantic space. Vi-LAD Elnoor et al. (2025), on the other hand, incorporates contrastive loss into the distillation process to explicitly enhance the discriminative power of the student model's attention distribution, encouraging it to focus on semantically key regions.

While these approaches have demonstrated strong performance in tasks such as image-text retrieval and visual question answering, they have yet to be systematically validated in UAV-VLN, particularly under resource-limited conditions. To address this gap, we propose the **DAD-SFT** framework, which utilizes *Cross-Modal Attention Distillation* to transfer the cross-modal attention distribution of the teacher model, thereby enhancing the student's multimodal perception. Simultaneously, we

introduce *Contrastive Attention Alignment*, which constructs diverse positive and negative samples to explicitly strengthen the model's ability to discriminate and generalize. This method achieves a balance between accuracy and efficiency, significantly outperforming existing approaches while maintaining lightweight model characteristics, and thus demonstrates strong potential for deployment in real-world UAV-VLN applications.

## C  PROMPTS TEMPLATE

---

**Prompt 1: UAV VLN task**

**System Message**:
You are an intelligent autonomous aerial vehicle (UAV) capable of real-world navigation and visual target localization.

**Mission Objective**:
Your mission is to locate a specific target described in natural language instructions.

**Details of the Targe**:
{target description}

**Environmental Perception**:
- The UAV's current position is indicated by the starting point of an arrow in the image, with its heading angle represented by the arrow's direction.
- The yellow box outlines the UAV's current camera field of view on the map, centered at pixel coordinates: cur_pose = {UAV current position}.
- Landmark regions are highlighted with red masks.

**Operational Guidance**:
- The target is usually located near a red-masked landmark.
- Use both the target description and the visual input to identify the most relevant red-masked landmark region.
- Infer the relative position of the target with respect to that landmark.

**Output Format Specification**:
- Present your reasoning process within <think> and </think> tags.
- Provide your final answer within <answer> and </answer> tags in the following format: {"target_location": [x, y]}
Your reasoning may include:
- A semantic interpretation of the target description.
- Identification of the correct landmark region.
- The bounding box of that region in the following format:
{"landmark_bbox": [x1, y1, x2, y2]}

---

## D  CROSS-MODAL ATTENTION DISTILLATION LAYER SELECTION ABLATION

To empirically validate the effectiveness of selecting only the *first-layer* cross-modal attention for distillation, we conduct an ablation study comparing two variants:

- **First-layer CAD**: Only the first-layer cross-modal attention is used for distillation.

- **Last-layer CAD**: Only the last-layer cross-modal attention is used for distillation.

All other settings (architecture, loss weights, optimizer, batch size, etc.) are kept identical. The performance is reported in Table 2.

We observe that distilling from the first-layer attention consistently outperforms distilling from the last-layer in all splits and across all four metrics. This supports our intuition that early-layer attention captures more transferable and generalizable cross-modal focus patterns, while deeper layers tend to encode more task-specific reasoning, which may not generalize well during distillation. These

Table 2: Ablation study comparing the effect of distilling attention from different layers. Distilling the first-layer attention consistently achieves better performance than the last-layer variant.

| Method | Validation Seen | | | | Validation Unseen | | | | Test Unseen | | | |
|---|---|---|---|---|---|---|---|---|---|---|---|---|
| | NE↓ | SR↑ | OSR↑ | SPL↑ | NE↓ | SR↑ | OSR↑ | SPL↑ | NE↓ | SR↑ | OSR↑ | SPL↑ |
| Last-layer CAD | 102.37 | 7.45 | 18.90 | 6.81 | 108.84 | 6.19 | 17.31 | 5.87 | 117.41 | 7.02 | 17.56 | 6.14 |
| First-layer CAD | **97.84** | **9.33** | **20.25** | **9.26** | **101.94** | **8.09** | **19.95** | **7.18** | **115.86** | **8.03** | **18.07** | **7.24** |

findings align with prior studies Feng et al. (2025); Chen et al. (2025) and further justify our design choice.

# E    PROMPTS TEMPLATE

---
**Prompt 2: Prompt for Adversarial Instruction Generation**

You are given a detailed description of the destination for the drone to locate or move toward the target area. Rewrite the description so that:

1. The rewritten description should remain almost identical in structure and details, except for changing one or two key attributes (e.g., color, side of the road, direction, relative position).
2. The modification should make the meaning conflict with the original (e.g., left and right, front and back, present and absent).
3. Do NOT change unrelated properties such as object type, scene background, or the general sentence structure.
4. Keep the rewritten sentence fluent and natural.

**Input description**:
```
{target description}
```

**Output description**:

---

# F    EVALUATION METRICS

- **NE (Navigation Error, ↓)**: The average distance between the UAV's final position and the target position.

- **SR (Success Rate, ↑)**: The proportion of episodes where the final position falls within a pre-defined threshold (20 meters) of the target.

- **OSR (Oracle Success Rate, ↑)**: Whether there exists at least one point along the trajectory where the UAV is within the success threshold.

- **SPL (Success weighted by Path Length, ↑)**: A metric that jointly considers navigation success and path efficiency.

# G    BASELINES

We provide brief descriptions of the baseline methods used in our experiments:

- **Random**
  At each time step, this baseline randomly samples an action without relying on any visual or textual input.

- **Sequence-to-Sequence (Seq2Seq)** Anderson et al. (2018)
  This model adopts a recurrent neural network architecture to encode visual observations and language instructions into a unified representation, which is then decoded into navigation outputs.

- **Cross-Modal Attention (CMA)** Liu et al. (2023)
  Based on the Seq2Seq, the CMA model introduces a cross-modal attention mechanism between vision and text, enhancing alignment between modalities and allowing the model to better focus on goals and landmarks described in the instructions.

- **Map-based Goal Predictor (MGP)** Lee et al. (2024)
  The MGP combines a language parser and an object detection module to extract goals and landmarks from the input, constructs a semantic map, and predicts the target location accordingly.

- **LLaMA-3.2-11B-Vision** Grattafiori et al. (2024)
  This is a VLM released by Meta, built upon the LLaMA architecture. It supports joint understanding and reasoning over visual and textual inputs, with 11 billion parameters and the capacity to handle complex multimodal tasks.

- **GPT-4o** OpenAI et al. (2024)
  GPT-4o is a VLM released by OpenAI, capable of simultaneously processing visual and textual inputs. It demonstrates strong cross-modal perception and reasoning abilities, achieving impressive performance on tasks involving instruction following and visual-language understanding.

- **Qwen2.5-VL Series** Bai et al. (2025)
  This series of VLMs is developed by Alibaba and includes models of different scales, designed to accommodate diverse application scenarios and resource constraints.

  - **Qwen2.5-VL-3B**: A lightweight version designed for high efficiency and fast inference. It is well-suited for deployment in resource-constrained environments and serves as the student model in this work.
  - **Qwen2.5-VL-7B**: A medium-scale model with enhanced capabilities in multimodal understanding and generation.
  - **Qwen2.5-VL-32B**: A high-performance version with powerful multimodal understanding and reasoning capabilities. It excels at handling complex visual-language tasks and serves as the teacher model in this work.

- **Naive RL**
  This baseline is trained purely under a reinforcement learning (RL) paradigm using the Group Relative Policy Optimization (GRPO) algorithm, where navigation policies are optimized through interaction with the environment. The reward function incorporates both **Goal Accuracy Reward**, based on the Euclidean distance between the predicted and ground-truth location, and **Format Reward**, which penalizes outputs that do not conform to required formats (e.g., invalid coordinates or malformed outputs). The training configuration is shown in Table 3.

Table 3: Hyperparameter configurations for Naive RL baseline.

| Hyperparameter | Value |
|---|---|
| Learning Rate | 1e-5 |
| Number of Epochs | 1 |
| Rollout per Step | 4 |
| LoRA Rank | 64 |

## H  HYPERPARAMETER SETTINGS

**Hyperparameter Justification.** For the loss weights $\lambda_{\text{attn}}$ and $\lambda_{\text{contrast}}$, our selection strategy was to fix all other hyperparameters and evaluate different values on the validation set. Specifically, we aimed for the auxiliary losses introduced by CAD and CAA to remain on the same order of magnitude as the main task loss (Task Loss), ensuring that they have sufficient influence during optimization without dominating the training process. Empirical results showed that setting $\lambda_{\text{attn}} = 50$ and $\lambda_{\text{contrast}} = 0.2$ achieves a favorable balance between accuracy and training stability.

The positive-to-negative sample ratio (1:64) was determined by following common practices in contrastive learning, where a large and diverse pool of negative samples helps the model learn clearer

Table 4: Hyperparameter configurations for DAD-SFT.

| Category | Hyperparameter | Value |
|---|---|---|
| Basic Training | Batch Size | 2 |
| | Learning Rate | 5e-6 |
| | Number of Epochs | 2 |
| | Optimizer | AdamW |
| Cross-Modal Attention Distillation | $\lambda_{\text{attn}}$ | 50 |
| Contrastive Attention Alignment | $\lambda_{\text{contrast}}$ | 0.2 |
| | Positive-to-Negative Sample Ratio | 1:64 |
| | InfoNCE Temperature Coefficient $\tau$ | 0.05 |

boundaries between *attended* and *non-attended* regions in the attention distribution. The temperature coefficient $\tau = 0.05$ controls the sensitivity of the similarity distribution in the InfoNCE loss, and this value was found to achieve favorable stability and discriminative ability in our experiments.

# I  EFFICIENCY EVALUATION RESULTS

Table 5: Comparison of resource consumption and inference latency across models on real devices.

| Model Name | Test Device | GPU Memory Usage (GB) | Inference Latency (s/step) |
|---|---|---|---|
| Seq2Seq | RTX 4090×1 | 0.92 | 1.00 |
| CMA | RTX 4090×1 | 0.87 | 1.01 |
| MGP | RTX 4090×1 | 0.84 | 0.82 |
| LLaMA-3.2-11B-Vision | RTX 4090×1 | 21.65 | 11.11 |
| GPT-4o | Cloud | N/A | 9.73 |
| Ours (3B) | RTX 4090×1 | 13.53 | 6.52 |
| Qwen2.5-VL-7B | RTX 4090×1 | 21.71 | 9.37 |
| Qwen2.5-VL-32B | A100 80GB×1 | 70.12 | 53.42 |

# J  CoT DATA ANALYSIS

To better understand the quality and characteristics of our constructed CoT training data, we conducted a systematic statistical analysis across multiple dimensions:

1. **Diversity of Instructions and Reasoning Trajectories**: We plotted histograms for both instruction lengths and CoT reasoning lengths (see Fig. 3(a) and Fig. 3(b)), showing significant variability. The instructions range from concise descriptions to complex multi-clause expressions, and the reasoning chains vary in length, reflecting diverse semantic structures and task difficulties. This ensures broad linguistic coverage.

2. **Spatial Distribution and Scale Balance of Landmark Bounding Boxes**: We calculated the sizes of all annotated landmark bounding boxes (Fig. 3(c)) and visualized the distribution of their center points across the map (Fig. 3(d)). The results show that landmarks are spatially well-distributed, and their sizes are reasonably balanced, indicating no significant annotation bias.

3. **Coverage of Target Coordinates**: The heatmap of target coordinates (Fig. 3(e)) demonstrates broad spatial coverage over the map. This indicates that the dataset encourages learning across diverse spatial contexts, minimizing risk of overfitting to localized regions.

4. **Semantic Alignment Between Landmarks and Targets**: We computed the spatial distances between each landmark center and the corresponding ground-truth target coordinate (Fig. 3(f)). The average distance is approximately 20.7 meters, confirming that landmarks mentioned in reasoning chains are meaningfully and spatially aligned with the final targets.

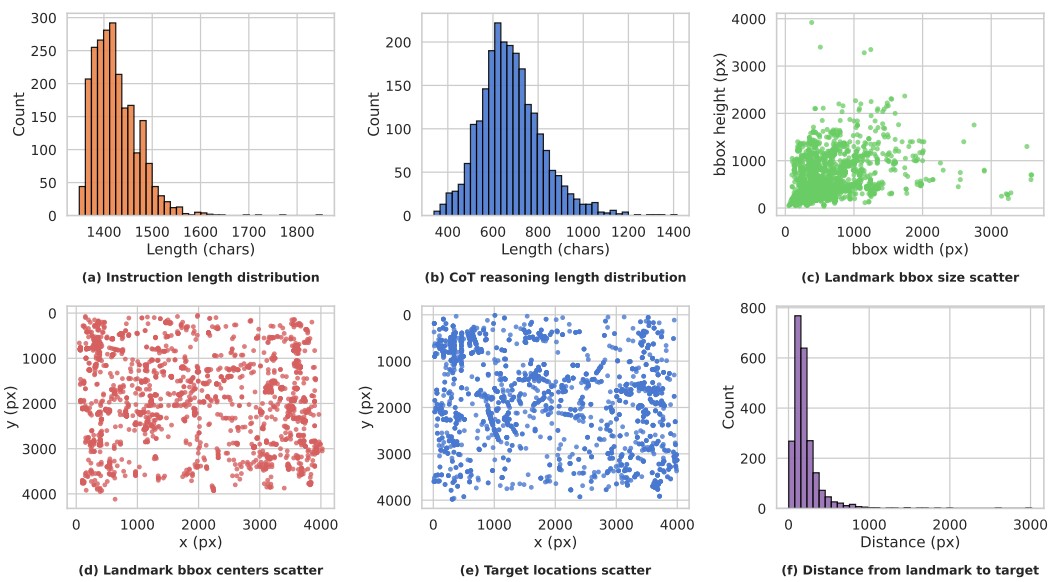

Figure 3: Statistical analysis of the CoT training data.

# K ADDITIONAL ABLATION STUDIES

## K.1 GRID SEARCH SENSITIVITY ANALYSIS ON $\lambda_{\text{ATTN}}$ AND $\lambda_{\text{CONTRAST}}$

We perform a sensitivity analysis of the loss weights $\lambda_{\text{attn}}$ and $\lambda_{\text{contrast}}$ on the Validation Unseen split. Table 6 shows the SR results under various combinations.

The results show that the configuration ($\lambda_{\text{attn}} = 50$, $\lambda_{\text{contrast}} = 0.2$) achieves the best performance. The model is more sensitive to $\lambda_{\text{attn}}$: when $\lambda_{\text{attn}}$ is too small (e.g., 10), the model underperforms, whereas a moderate range (50–80) yields more stable and superior results. This suggests that attention distillation loss is more effective when applied with sufficient strength. In contrast, the model is relatively robust to $\lambda_{\text{contrast}}$ within a reasonable range, with particularly strong stability observed between 0.1 and 0.2.

Table 6: SR (%) on Validation Unseen under different $\lambda_{\text{attn}}$ and $\lambda_{\text{contrast}}$ combinations.

| $\lambda_{\text{attn}} \setminus \lambda_{\text{contrast}}$ | 0.05 | 0.1 | 0.2 | 0.5 |
|---|---|---|---|---|
| 10 | 7.85 | 8.12 | 8.46 | 7.90 |
| 30 | 9.02 | 9.56 | 10.18 | 9.61 |
| 50 | 10.12 | 10.26 | **10.43** | 10.01 |
| 80 | 9.76 | 10.07 | 10.22 | 9.88 |

## K.2 CONTRIBUTION ABLATION OF DIFFERENT NEGATIVE SAMPLE TYPES

To understand the contribution of different negative sample types in CAA, we conduct ablation experiments under the CAA-only setting (i.e., without CAD). The results on the Test Unseen split are presented in Table 7.

The results indicate that all negative sample types positively contribute to performance. Notably, Adversarial Attention and Cross-instance Attention negatives are most impactful in improving the model's discriminability and overall performance. Removing any of them leads to a performance drop, validating the necessity of diverse negative designs in CAA.

Table 7: Effect of different negative sample types in CAA (SR %) on Test Unseen.

| Configuration | SR (%) |
|---|---|
| Naive SFT | 5.79 |
| **CAA-only (full negative types)** | **9.79** |
| CAA-only w/o Random Attention Negatives | 9.12 |
| CAA-only w/o Perturbed Attention Negatives | 9.26 |
| CAA-only w/o Adversarial Attention Negatives | 8.87 |
| CAA-only w/o Cross-instance Attention Negatives | 9.03 |

