# OpenReview forum: "DAD-SFT: Dual Attention Distillation for Lightweight UAV Vision-Language Navigation"
_ICLR.cc/2026/Conference — Submitted to ICLR 2026_

### Official Review · Reviewer_wPUD · 2025-10-25

**Soundness:** 2
**Presentation:** 3
**Contribution:** 3
**Rating:** 4
**Confidence:** 5

**Summary:**

This paper proposes a vision-language model (VLM) to tackle aerial vision-language navigation tasks.  VLMs have demonstrated strong cross-modal understanding and reasoning.  However, these capabilities scale with the model size while larger VLMs have heavier computational overhead, making them inefficient for real-time decision.  This paper addresses such limitations by distilling large VLMs into compact student models based on the guidance of cross-modal attention maps, where the student models learn to generate similar attention maps as the teacher model.  To enhance robustness, this paper additioanlly proposes a contrastive attention objective with the introduction of negative attention maps.  This paper demonstrates that jointly training with task loss, attention distillation loss and contrastive attention loss makes the student model as performant as the teacher model.

**Strengths:**

1. This paper is well written.  Readers can easily follow the rationale and the implementation details of the proposed method.
2. This paper demonstrates strong performance on CityNav dataset, outperfroming other baselines by a large margin.
3. This paper presents detailed ablation study, demonstrating the efficacy of each model component
4. While the contrastive learning idea is commonly known, combining attention-based distillation with contrastive learning seems novel.

**Weaknesses:**

1. This paper considers navigation actions as texts.  However, texts  are not suitable to represent high-precision floating numbers, required for robotic navigation tasks.
2. Following Weakness 1, this paper does not consider a strong baseline--OpenVLA [1] / SpatialVLA [2] and $\pi_0$ [3], as the former predict discretized action tokens and the latter predicts continual action control.  These models should have similar model size (7B) as the student model used in this paper, meanwhile they have shown strong performance in robotic manipulation tasks.  They should be considered as strong baselines for navigation tasks and compared against the proposed method.
3. This paper lacks discussion of recent studies on training-free aerial navigation policies.  See-Point-Fly [4] proposes that navigation tasks are inherent visual grounding tasks in static scenes.  Since VLMs excel at visual grounding, one can directly use VLM models to generate 2D waypoints for 3D navigation, without the need for training navigation policies.  I'd strongly encourage the authors to include discussions on these new perspectives.
4. This paper only conducts experiments on a single simulation benchmark, which seems to be relatively small-scale.  I'd highly encourage the authors to evaluate the proposed method on a larger-scale simulation benchmark--OpenNAV[5], to showcase the robustness of the proposed method.

---

Reference:

[1] Kim, Moo Jin, et al. "Openvla: An open-source vision-language-action model." arXiv preprint arXiv:2406.09246 (2024).

[2] Qu, Delin, et al. "Spatialvla: Exploring spatial representations for visual-language-action model." arXiv preprint arXiv:2501.15830 (2025).

[3] Black, Kevin, et al. "$\pi_0 $: A Vision-Language-Action Flow Model for General Robot Control." arXiv preprint arXiv:2410.24164 (2024).

[4] Hu, Chih Yao, et al. "See, Point, Fly: A Learning-Free VLM Framework for Universal Unmanned Aerial Navigation." Conference on Robot Learning. PMLR, 2025.

[5] Qiao, Yanyuan, et al. "Open-nav: Exploring zero-shot vision-and-language navigation in continuous environment with open-source llms." 2025 IEEE International Conference on Robotics and Automation (ICRA). IEEE, 2025.

**Questions:**

1. How does DAD-SFT compare to OpenVLA, SpatialVLA and $\pi_0$ on CityNav?
2. Can you discuss recent studies that reformulate navigation tasks as visual grounding tasks?
3. How does DAD-SFT work on OpenNAV?

---

> ### Author Response · Authors · 2025-11-20
> **Q1: This paper considers navigation actions as texts. However, texts are not suitable to represent high-precision floating numbers, required for robotic navigation tasks.**
>
> Thank you for raising this important concern.
>
> The main reason we chose to output actions as text is because the CityNav benchmark task itself uses a discrete action space (e.g., move forward, turn left, turn right, stop) and does not require continuous control. This is consistent with the baseline methods and prior work such as Seq2Seq, CMA, MGP, FlightGPT, etc., all of which also use a discrete action modeling approach. We followed this task setup.
>
> Moreover, the design of our DAD-SFT framework can easily be extended to non-textual outputs, allowing the model to directly output specific machine control commands, including floating-point values. This would not alter our training framework because the CAD/CAA mechanisms primarily focus on perception alignment, decoupling it from the output format.

---

> ### Author Response · Authors · 2025-11-20
> **Q2: The paper does not compare with strong VLA models such as OpenVLA, SpatialVLA, $\pi_{0}$.**
>
> Thank you for pointing out these important recent works. We fully acknowledge the cutting-edge nature of OpenVLA, SpatialVLA, and $\pi_{0}$ in the VLA tasks, and their model sizes are comparable to our student model.
>
> We did not include them in our experiments primarily because: Most of these methods focus on robot manipulation or indoor vision navigation, and their input-output formats differ from CityNav. Directly using the public code, weights, or evaluation interfaces of these methods does not allow us to obtain their performance on CityNav. Adapting them to this task would require significant time and cost.
>
> We plan to expand our discussion of these works in the Related Work section of the revised version. If the review cycle allows, we will also attempt to adapt some of these models and report their performance on CityNav.

---

> ### Author Response · Authors · 2025-11-20
> **Q3: Lack of discussion on training-free visual grounding navigation methods, such as See-Point-Fly.**
>
> We appreciate the reviewer’s suggestion to discuss See-Point-Fly, an inspiring work. We agree with the core idea of training-free navigation: in static environments, the powerful visual understanding and localization capabilities of VLMs can be directly used to generate 2D waypoints, thus enabling navigation without additional training. See-Point-Fly reexamines the traditional "policy learning paradigm" and emphasizes that navigation tasks are essentially visual grounding problems. This perspective is highly valuable for UAV navigation research, particularly in scenarios where the target is visible and the description is clear.
>
> We acknowledge that our current version lacks sufficient discussion on this training-free paradigm. In the revised version, we will add a systematic discussion of such methods, especially See-Point-Fly, in the Related Work and Discussion sections, highlighting the differences in task modeling between it and our method. Moreover, in future work, we will consider incorporating See-Point-Fly as a baseline to more comprehensively evaluate the model's performance.

---

> ### Author Response · Authors · 2025-11-20
> **Q4: The experiments are only conducted on CityNav, which is relatively small in scale. It is suggested to test the model on larger-scale platforms such as OpenNAV to verify its generalization ability.**
>
> Thank you for raising the concern about the experiment scope. We chose CityNav as our primary test platform because it provides a city-scale simulation environment, a clear semantic structure, and a complete map input mechanism. Furthermore, several existing methods (such as Seq2Seq, CMA, MGP) have been evaluated on CityNav, which facilitates fair and direct performance comparison.
>
> OpenNAV is indeed a very challenging zero-shot navigation platform, and we believe it is extremely valuable. We highly recognize OpenNAV’s potential to provide a more complex and open evaluation environment, particularly for testing the generalization ability of the model.
>
> Since our system has not yet been adapted to the data interfaces and input structure of OpenNAV, we have not included corresponding experiments in this version. However, we have already identified this as a key future direction, and we plan to migrate DAD-SFT to OpenNAV in future versions for comprehensive evaluation.

---

> ### Author Response · Authors · 2025-11-20
> **Q5: Can you discuss recent studies that reformulate navigation tasks as visual grounding tasks?**
>
> Thank you for bringing up this key direction. In recent years, there has been an emerging trend of reformulating navigation tasks as "visual grounding + navigation execution" in visual-language navigation (VLN) research. These methods emphasize first determining the target location using language and vision understanding, followed by path planning and action execution. This represents a significant shift in task modeling. Below are two representative studies:
>
> - 《See, Point, Fly》: This work proposes a training-free UAV visual-language navigation framework. The model directly uses a visual-language model to generate 2D image waypoints based on natural language instructions, which are then mapped to 3D displacement vectors to complete navigation. This method treats navigation as a visual grounding problem and skips the traditional policy learning step.
>
> - 《FlightGPT》: This study focuses on the generalization ability and explainability in UAV VLN. It proposes a two-stage navigation process: first, the model predicts the target location coordinates based on language descriptions and environmental visual input using a visual-language model (VLM) with a Chain of Thought mechanism; then, a Look-ahead strategy is applied for path planning and action sequence generation.
>
> These studies reflect the trend of shifting navigation tasks from policy learning paradigms to a layered structure of language understanding, visual grounding, and action execution. We will add a new section titled "Navigation as Visual Grounding" in the Related Work section of the revised version to cover this direction.

---

> ### Author Response · Authors · 2025-11-20
> **Q6: How does DAD-SFT work on OpenNAV?**
>
> We highly appreciate the work of OpenNAV, which cleverly combines structured reasoning processes with multimodal perception modules, demonstrating the excellent potential of large-scale language models in continuous visual-language navigation tasks. While our DAD-SFT method has not been directly tested in the OpenNAV framework, its design is precisely aimed at addressing the resource bottlenecks and response efficiency issues that such systems face in real-world deployments.
>
> Specifically, the LLM navigator in OpenNAV is responsible for multiple-stage reasoning tasks, such as instruction understanding, progress evaluation, and action decision-making, and exhibits significant reasoning strength and cross-modal integration abilities. On this basis, DAD-SFT can naturally be integrated as a model compression solution: we can treat the strong LLM in OpenNAV (such as LLaMA3.1-70B) as the teacher model and use the CAD mechanism proposed in DAD-SFT to transfer its navigation capabilities to a smaller, faster student model.
>
> To maintain the generalization ability in navigation scenarios, we can also leverage the Contrastive Attention Alignment mechanism in DAD-SFT to strengthen the student model's ability to recognize key visual features and semantic correspondences in complex scenarios. This approach not only significantly reduces computational costs but also retains the interpretability of reasoning chains and spatial decision-making processes, making it suitable for deployment on edge devices or real-time control scenarios.
>
> Therefore, while DAD-SFT has not yet been applied directly to OpenNAV, it is highly compatible and holds great potential as a supporting solution for lightweight deployment in the future.

---

> > ### Comment · Reviewer_wPUD · 2025-11-23
> >
> > Thank the authors for the response.  My concerns of (1) using texts as the action representations, and (2) lacking discussion on recent work, are addressed.  However, the authors haven't presented comparison to either VLA models finetuned for aerial navigation or training-free navigation frameworks like SPF.  Meanwhile, the authors hanve't shown more results on other benchmark.  I'd highly recommend the authors to show some results before the end of discussion phase.

---

> > > ### Author Response · Authors · 2025-12-03
> > >
> > > We thank the reviewer for the continued engagement and constructive suggestions.
> > >
> > > 1. On the comparison with SPF baseline
> > > We have carefully analyzed the SPF (See, Point, Fly) framework and acknowledge that its core strengths lie in combining depth estimation and 3D geometric reasoning to enable training-free, closed-loop UAV control. In contrast, CityNav is a 2D navigation task, where such 3D spatial reasoning capabilities cannot be fully utilized. While a simplified 2D adaptation of SPF is technically feasible, it would deviate from its original design goals and fail to provide a representative or fair comparison.
> > >
> > > 2. On evaluation beyond CityNav
> > > We believe that CityNav, with its multiple splits (Seen / Unseen / Test Unseen), already provides a diverse set of environments and semantic combinations, allowing effective evaluation of generalization in complex urban settings. Therefore, we consider our current experimental setup to be both representative and informative. At the same time, we fully agree with the reviewer that evaluation on larger-scale or different types of platforms (e.g., OpenNAV, OpenFly) would further strengthen the conclusions. This will be a key direction of our future work.
> > >
> > > Once again, we thank the reviewer for the valuable feedback, which helps us further refine our research direction and experimental scope.

---

### Official Review · Reviewer_3tX4 · 2025-10-27

**Soundness:** 3
**Presentation:** 3
**Contribution:** 3
**Rating:** 6
**Confidence:** 3

**Summary:**

This paper addresses the deployment of large Vision-Language Models for UAV navigation on resource-constrained devices. The authors propose DAD-SFT, which distills knowledge from a 32B teacher to a 3B student model through: (1) Cross-Modal Attention Distillation (CAD) that transfers first-layer attention patterns via KL divergence, and (2) Contrastive Attention Alignment (CAA) that uses teacher attention as positive samples with four negative types (random, perturbed, adversarial, cross-instance) optimized via InfoNCE loss. On CityNav benchmark, the 3B student matches or exceeds the 32B teacher's performance (12.96% vs 11.98% SR on Test Unseen) while using 5× less memory (13.53GB vs 70.12GB) and 8× faster inference (6.52s vs 53.42s), enabling single RTX 4090 deployment. Ablations confirm that both mechanisms contribute complementarily to achieving efficient yet accurate UAV navigation.

**Strengths:**

- The paper addresses a genuine deployment challenge in robotics/UAV applications: powerful VLMs cannot run on resource-constrained devices.
- The attention-level distillation approach is conceptually elegant. Rather than distilling outputs or deep features, the method transfers "where to look" patterns from the teacher. This is intuitive for navigation tasks that require spatial grounding.
- The decision to use first-layer attention (rather than deeper layers) is supported by both prior literature and empirical ablations (Table 2), which show that shallow layers capture more transferable cross-modal correspondences.

**Weaknesses:**

- The paper evaluates only on the CityNav dataset within a single simulation environment. For claims about "cross-scene generalization" and practical UAV deployment, this is insufficient. The "unseen" splits are still part of the same city-scale environment, with similar visual characteristics. No evaluation on other VLN benchmarks (R2R [A], REVERIE [B], SOON [C]), different simulation environments, or real-world data undermines the generalization claims.
- The CoT training data (2,300 samples) is synthetically generated by the teacher model (Qwen2.5-VL-32B), filtered to remove errors >20m, and then the predicted coordinates are replaced with ground truth. This raises several concerns: (a) the student learns from the same model family it distills from, potentially amplifying biases, (b) replacing predictions with ground truth creates a mismatch between reasoning and answers, (c) only 2,300 samples seems small for training a 3B model, and (d) no analysis of data quality or potential distribution shifts is provided.
- Both teacher (Qwen2.5-VL-32B) and student (Qwen2.5-VL-3B) share identical architectures from the same model family. This raises critical questions: Does attention distillation work across different architectures? The paper claims this is a "unified framework," but provides no evidence that it generalizes beyond Qwen2.5-VL variants.
- The paper lacks comparisons with established knowledge distillation methods (FitNet, Attention Transfer, RelationKD) or other attention distillation approaches mentioned in related work (Feng et al. 2025 [D], Elnoor et al. 2025 [E], Zhou et al. 2025 [F]). Also, there is another trend of using MLLMs/VLMs to perform VLN in a zero-shot manner (SPF [G] and OpenFly [H]), which is worth comparing. The "Naive RL" baseline is poorly explained (only mentioned in one sentence in Section 4.2).

[A] Weihs, Luca, et al. "Visual room rearrangement." Proceedings of the IEEE/CVF conference on computer vision and pattern recognition. 2021.

[B] Qi, Yuankai, et al. "Reverie: Remote embodied visual referring expression in real indoor environments." Proceedings of the IEEE/CVF Conference on Computer Vision and Pattern Recognition. 2020.

[C] Zhu, Fengda, et al. "Soon: Scenario oriented object navigation with graph-based exploration." Proceedings of the IEEE/CVF Conference on Computer Vision and Pattern Recognition. 2021.

[D] Ge, Yuyao, et al. "Focusing by Contrastive Attention: Enhancing VLMs' Visual Reasoning." arXiv preprint arXiv:2509.06461 (2025).

[E] Elnoor, Mohamed, et al. "Vi-LAD: Vision-Language Attention Distillation for Socially-Aware Robot Navigation in Dynamic Environments." arXiv preprint arXiv:2503.09820 (2025).

[F] Zhou, Yang, et al. "Attention distillation: A unified approach to visual characteristics transfer." Proceedings of the Computer Vision and Pattern Recognition Conference. 2025.

[G] Hu, Chih Yao, et al. "See, Point, Fly: A Learning-Free VLM Framework for Universal Unmanned Aerial Navigation." Conference on Robot Learning. PMLR, 2025.

[H] Gao, Yunpeng, et al. "OpenFly: A Comprehensive Platform for Aerial Vision-Language Navigation." arXiv preprint arXiv:2502.18041 (2025).

**Questions:**

- Can you provide results on other VLN benchmarks (R2R, REVERIE, SOON), or on other UAV datasets mentioned in the related work, such as the dataset from Wang et al. 2024? Even preliminary results would help assess whether the method generalizes beyond the specific CityNav environment. If not available, can you explain why CityNav alone is sufficient to validate your claims about "cross-scene generalization"?
- What is the impact of your data construction procedure? Specifically: (a) What happens if you use ground-truth reasoning from human annotations instead of teacher-generated CoT? (b) How does performance change if you don't replace predicted coordinates with ground truth? (c) Can you provide an analysis showing the teacher's reasoning quality and how the filtering affects data distribution? (d) Why are 2,300 samples sufficient for training a 3B model?
- Can you demonstrate that DAD-SFT works when the teacher and student have different architectures (e.g., LLaMA-3.2-11B as the teacher and Qwen2.5-VL-3B as the student, or vice versa)?
- How do you explain the 3B student achieving higher performance than the 32B teacher on Test Unseen (12.96% vs 11.98% SR)? Did you run multiple seeds and compute confidence intervals? Could this be overfitting, evaluation inconsistency, or statistical noise?

---

> ### Author Response · Authors · 2025-11-20
> **Q1: The dataset scope is limited to CityNav, which does not support claims of “cross-scene generalization.”**
>
> Thank you for pointing out the limitations of using only the CityNav dataset for generalization evaluation.
>
>  In this work, we chose to conduct our evaluation solely on the CityNav dataset because it features realistic maps, rich semantic annotations, and standardized evaluation protocols, making it one of the most representative benchmarks for UAV VLN in urban environments.
>
> The generalization capability of DAD-SFT is primarily reflected in its performance across unseen regions within the city environment. Specifically, our model maintains consistently strong performance on the Validation Unseen and Test Unseen splits, even when facing diverse combinations of maps, landmark distributions, language styles, and navigation targets. This demonstrates the model's ability to adapt to complex semantic variations in unseen urban regions.
>
> We fully agree that relying on a single dataset is insufficient to fully support broader generalization claims. As such, we plan to include additional evaluations on multiple VLN benchmarks such as R2R, REVERIE, and SOON in future versions, to further validate the generality of our method.

---

> ### Author Response · Authors · 2025-11-20
> **Q2: Potential issues in the CoT data construction process, including model bias, small data size, and lack of quality analysis.**
>
> Thank you for your thoughtful concerns regarding our CoT data construction strategy. Our detailed responses are as follows:
>
> (a) Model bias
>
> We used Qwen2.5-VL-32B as the data generator based on its strong zero-shot performance and generation efficiency. We acknowledge that relying on a single model may introduce potential biases.
>
> We plan to explore generating CoT data using other models (e.g., GPT-4o, LLaMA-3.2) in future work to improve diversity and validate the robustness of our framework.
>
> (b) Answer replacement and reasoning alignment
>
> To reduce inconsistencies between reasoning chains and final answers, we adopt strict filtering during data generation: only samples with a predicted location error below 20 meters are retained (see Sec. 3.3.2). This ensures that the reasoning process aligns well with the ground-truth target area. Replacing the predicted answer with ground truth further improves training stability and prevents error propagation during learning.
>
> We also plan to conduct controlled comparisons between the replacement and non-replacement strategies in future work.
>
> (c) Small data size
>
> We acknowledge that 2,300 samples may seem limited, but this was a practical trade-off under time and resource constraints. Nevertheless, we ensured substantial data diversity by incorporating different regions, various landmarks, language styles, and navigation targets. A three-stage filtering pipeline was used to guarantee quality.
>
> Under this small dataset, DAD-SFT still yields significant performance gains (e.g., Test Unseen SR improves from 1.45% to 12.96%), suggesting that the framework is data-efficient and robust to limited training size.
>
> (d) Data Quality Analysis
>
> Thank you for raising concerns regarding data quality. In response, we have conducted a systematic analysis of the CoT training data and included the corresponding statistical charts in Section 3.3.2 and Appendix J (Cot Data Analysis). The main findings are as follows:
>
> 1. Diversity of Instructions and Reasoning Trajectories
>
> We plotted histograms for instruction length and CoT reasoning length (see Fig. 6(a) and Fig. 6(b)), showing significant variability in both. This indicates that our data includes a wide range of inputs, from concise instructions to complex scene descriptions, with varying lengths of reasoning processes. This reflects the different levels of task difficulty and semantic structures, ensuring linguistic diversity and comprehensive coverage in the dataset.
>
> 2. Spatial Distribution and Scale Balance of Landmark Bounding Boxes
>
>  We calculated the sizes of all landmark bounding boxes (see Fig. 6 (c)) and plotted a scatter diagram of their center positions (see Fig. 6 (d)). The results show that the landmarks are distributed relatively evenly across the map, with no significant spatial bias. This indicates that the landmark annotations maintain a good balance in size and are spatially representative.
>
> 3. Coverage of Target Coordinates
>
>  The spatial distribution of target coordinates is shown in Fig. 6(e), demonstrating good coverage across the entire map. This indicates that the navigation targets span a diverse range of spatial regions, mitigating the risk of biased training data.
>
> 4. Semantic Alignment Between Landmarks and Targets
>
> We further calculated the spatial distance between the center of the landmark bounding box and the ground-truth target coordinates (see Fig. 6(f)), with an average distance of approximately 20.7 meters. This suggests that the landmark region referenced by the model is well-aligned with the final target location, confirming the effectiveness of the semantic alignment in the reasoning process.

---

> ### Author Response · Authors · 2025-11-20
> **Q3: Teacher and student models share the same architecture—can DAD-SFT generalize across different model architecture?**
>
> Thank you for raising this important point. We agree that validating DAD-SFT across different model architectures is crucial for demonstrating its generality.
>
> We would like to clarify that the current DAD-SFT framework requires alignment of attention modules, especially in the Cross-Modal Attention Distillation (CAD). This alignment is only feasible when the teacher and student share compatible attention structures. As a result, DAD-SFT currently supports intra-family distillation (e.g., Qwen → Qwen or LLaMA → LLaMA), but not cross-family setups (e.g., LLaMA → Qwen).
>
> We are actively conducting additional experiments using the LLaMA model family (e.g., LLaMA-3.2-11B as teacher and LLaMA-3.2-3B as student) to evaluate the effectiveness of DAD-SFT in a different architecture setting.

---

> ### Author Response · Authors · 2025-11-20
> **Q4: Lack of comparisons with established knowledge distillation and attention distillation methods.**
>
> Thank you for pointing this out. We acknowledge the current manuscript does not include direct comparisons with classic or recent distillation approaches. In future research, we plan to incorporate the following comparisons:
>
> - Classic methods: FitNet, Attention Transfer, and RelationKD.
> - Recent attention distillation approaches: Vi-LAD, Attention distillation.
> - Zero-shot VLN baselines: SPF and OpenFly.
>
> These comparisons will further validate the effectiveness of DAD-SFT under various supervision paradigms.

---

> ### Author Response · Authors · 2025-11-20
> **Q5: The “Naive RL” baseline is poorly explained.**
>
> Thank you for highlighting this issue. We agree that the description in the paper was overly brief. Here we provide further details.
>
> The Naive RL baseline is trained entirely using a reinforcement learning paradigm based on the Group Relative Policy Optimization (GRPO) algorithm. The agent learns goal prediction by interacting with the environment. The reward function includes:
>
> - Goal Accuracy Reward: Rewards based on the Euclidean distance between the predicted and ground-truth location.
> - Format Reward: Penalizes outputs that do not conform to required formats (e.g., invalid coordinates or malformed outputs).
> The model uses the same lightweight architecture (Qwen2.5-VL-3B) as our student model to provide a fair comparison.
>
> Training configuration:
> | **Learning Rate** | **Epochs** | **Rollout** | **LoRA-r** |
> |--------------------|------------|-------------|------------|
> | 1e-5               | 1          | 4           | 64         |
>
> We have included these details in Appendix G (Baselines) of the revised version to improve reproducibility and clarity.

---

> ### Author Response · Authors · 2025-11-20
> **Q6: Why does the student outperform the teacher? Could this be due to statistical noise? Were multiple seeds tested?**
>
> This is an important question—thank you for raising it. Our analysis is as follows:
>
> 1. High-quality CoT supervision:
>  The student benefits from learning on carefully filtered, teacher-generated CoT data. This provides rich intermediate reasoning supervision that helps the student model learn complex semantic grounding more effectively.
>
> 2. Contrastive Attention Alignment (CAA):
>  The CAA module enhances the student’s ability to distinguish relevant from irrelevant regions via negative samples. The teacher model, in contrast, is not trained with this mechanism, making it more susceptible to semantic shifts in unseen environments.
>
> 3. Task-specific optimization:
>  While the teacher is a general-purpose model trained across diverse tasks (e.g., VQA, captioning, open-ended chat), the student is trained under DAD-SFT with supervision tailored specifically for navigation and goal localization. This focused training improves adaptation to the VLN task.
>
> To assess the robustness of this improvement, we conducted training and evaluation under multiple random seeds. Performance on the Test-Unseen split is summarized below:
>
> | Seed     | NE (↓) | SR (↑) | OSR (↑) | SPL (↑) |
> |----------|--------|--------|---------|---------|
> | Seed A (Original) | 76.94  | 12.96  | 26.78   | 12.02   |
> | Seed B            | 75.82  | 13.05  | 26.49   | 12.16   |
> | Seed C            | 78.33  | 12.81  | 26.32   | 11.90   |
> | **Mean**          | 77.03  | 12.94  | 26.53   | 12.03   |
>
> The results show that the model's performance varies very little across different seeds, with the success rate (SR) consistently stable between 12.8% and 13.1%. Other metrics, such as NE, SPL, and OSR, also fluctuate within a narrow range. This indicates that the performance improvement brought by DAD-SFT is statistically robust and not the result of random variations or specific training conditions.
>
> We will include this table and accompanying discussion in the revised version to strengthen the reproducibility and credibility of our results.

---

> > ### Comment · Reviewer_3tX4 · 2025-11-23
> > **comment**
> >
> > Thank the authors for the detailed rebuttal and additional analysis. I think most of my concerns regarding performance robustness, data quality (in Appendix J), and baseline details are resolved. However, the paper still relies only on the CityNav dataset for evaluation (although this is not the limitation of this paper but rather a limitation in this field). Also, there are still no comparisons with FitNet, Vi-LAD, SPF, or OpenFly. I still hope to see the comparisons with these baseline methods before the discussion phase ends (not sure if this is possible) or in the future.

---

> > > ### Comment · Reviewer_3tX4 · 2025-11-26
> > > **Will additional evaluations be available?**
> > >
> > > Could the authors comment on whether the additional comparisons/evaluations with missing baseline methods like FitNet, Vi-LAD, SPF, or OpenFly will be available before the discussion phase ends?
> > > If these evaluations are not valid, I am curious about what stops the authors from comparing against these methods. Is it because these methods take a long time to run? Or are there many scenes that need to be evaluated?

---

> > > > ### Author Response · Authors · 2025-11-27
> > > > **Response on the Feasibility of Incorporating Vi-LAD as a Baseline**
> > > >
> > > > Thank you for pointing out additional baseline methods for consideration. We are in the process of thoroughly evaluating each of them and will respond to their applicability one by one.
> > > >
> > > > The following is our response regarding Vi-LAD:
> > > >
> > > > We have carefully studied the Vi-LAD methodology. To make a fair and technically valid comparison within our UAV-VLN setting, a direct adaptation of Vi-LAD would require:
> > > >
> > > > 1. Dual-teacher distillation, involving both a vision-action navigation model (e.g., VANP) and a large VLM;
> > > >
> > > > 2. Integration with a Model Predictive Control (MPC) framework for continuous trajectory optimization.
> > > >
> > > > However, these adaptations are non-trivial in our setting:
> > > >
> > > > - In UAV-VLN, we currently lack a suitable open-source vision-action model (like VANP) that supports aerial semantic reasoning and can serve as a second teacher;
> > > >
> > > > - Transforming our discrete decision pipeline into an MPC-based control loop would require significant architectural redesign and re-engineering, including trajectory-level supervision and continuous control integration.
> > > >
> > > > Given the technical gap and development effort, we are unable to complete such a comparison within the discussion phase.

---

> > > > > ### Comment · Reviewer_3tX4 · 2025-11-27
> > > > > **other baselines?**
> > > > >
> > > > > Thank the authors for the clarification about Vi-LAD. I now understand why the comparison is not straightforward, as Vi-LAD relies on MPC. I suggest the authors note this difficulty specifically in the revised paper.
> > > > >
> > > > > What about other baselines like FitNet, SPF, and OpenFly? Do they encounter similar challenges? Even the evaluation on a subset within the discussion phase will be appreciated.

---

> > > > > > ### Author Response · Authors · 2025-11-28
> > > > > >
> > > > > > Thank you for your understanding and helpful suggestion. We will make sure to explicitly mention the adaptation challenges of Vi-LAD—particularly its reliance on dual-teacher distillation and MPC—in the revised version of the paper.

---

> > > > ### Author Response · Authors · 2025-11-28
> > > > **Response on the Feasibility of Incorporating FitNets as a Baseline**
> > > >
> > > > We have carefully studied the training strategy proposed in FitNets, and attempted to adapt its core ideas to our UAV-VLN task. The resulting design is as follows:
> > > >
> > > > | Stage         | Mimic Target                                  | Optimization Objective                                                | Parameters to Update                     |
> > > > |---------------|-----------------------------------------------|------------------------------------------------------------------------|------------------------------------------|
> > > > | Stage 1 (Hint) | Student mid-layer attention → Teacher mid-layer attention | L_hint = KL(Attention_teacher, Attention_student)                                    | Student lower g layers |
> > > > | Stage 2 (Knowledge Distil)   | Output → labels + teacher soft output        | L_total = L_task + lambda_soft * L_soft-task                                       | All student parameters (End-to-End)      |
> > > >
> > > >
> > > > In Stage 1, we introduce the teacher model’s intermediate cross-modal attention as a hint signal, guiding the student model to form attention distributions that resemble those of the teacher in the corresponding guided layer. KL divergence is used to align the distributions.
> > > >
> > > > Stage 2 then proceeds with end-to-end training, optimizing a combined objective that includes ground-truth labels and soft targets from the teacher model.
> > > >
> > > > While this design is technically sound and well-motivated, completing the full implementation and experimental validation requires additional time. We are actively working on this integration and will make every effort to finish the corresponding experiments before the discussion phase concludes.

---

> > > > > ### Author Response · Authors · 2025-12-02
> > > > > **Preliminary Results and Analysis of the FitNetsBaseline**
> > > > >
> > > > > Following the training philosophy of FitNets, we designed a two-stage distillation pipeline tailored to the UAV-VLN task. The first stage performs mid-layer cross-modal attention alignment (Hint), and the second stage incorporates soft-label supervision (Knowledge Distillation). We have completed preliminary experiments on the CityNav dataset under the Test Unseen split, and the result is:
> > > > >
> > > > > Qwen2.5-VL-3B (FitNets KD): SR = 8.12%
> > > > >
> > > > > The result is slightly higher than that of CAD-only (8.03%), suggesting that, in addition to attention distillation, introducing soft outputs from the teacher provides richer supervision and helps the student model better capture semantic cues. These soft labels offer more nuanced guidance in cases where the target description is ambiguous or the scene is complex, which helps mitigate overfitting to hard labels.
> > > > >
> > > > > However, the method still underperforms CAA-only (9.79%) and our full DAD-SFT framework (12.96%). We believe the main limitation is that the FitNets framework lacks mechanisms to explicitly suppress incorrect attention patterns. In contrast, CAA constructs diverse negative samples, effectively distinguishing “what should be attended to” from “what should not,” thereby strengthening the model’s discriminative ability and generalization. Furthermore, DAD-SFT integrates both CAD and CAA, and the two components complement each other in perceptual alignment and discriminative enhancement, leading to the best performance across all settings.
> > > > >
> > > > > We will include the complete FitNets results and analysis in the final submission.

---

> > > > ### Author Response · Authors · 2025-11-29
> > > > **Response on the Feasibility of Incorporating OpenFly as a Baseline**
> > > >
> > > > We have carefully analyzed the OpenFly approach, and we agree that it is feasible to adapt it to the CityNav dataset.
> > > >
> > > > Specifically, we plan to adopt the OpenFly-Agent architecture to build a keyframe-based multi-frame vision-language navigation model on CityNav. The migration involves several core steps: (1) extracting representative keyframes from the original CityNav trajectories using strategies such as detecting action transition points or semantic scene changes; (2) combining the current frame with selected keyframes to form a multi-frame observation sequence, which will be jointly encoded by the visual encoder; (3) fusing the natural language instruction with the visual representations to predict the appropriate action at each time step through a multimodal decision-making module.
> > > >
> > > > The main implementation effort lies in the following aspects:
> > > >
> > > >  (1) Transforming the CityNav data into the format required by OpenFly-Agent by implementing keyframe selection strategies and enabling support for multi-frame inputs, which also requires reconstructing the data loading pipeline;
> > > >
> > > > (2) Introducing a history-aware frame caching mechanism, as the current model operates under a single-frame input paradigm;
> > > >
> > > >  (3) Aligning the action space, since the action definitions in CityNav differ from those used in OpenFly—this requires mapping and adjusting the action categories and output format accordingly.
> > > >
> > > > We have already started implementing the above adaptations. However, as the migration involves systemic changes to both data structures and model architecture, completing the full experimental pipeline will take some time. We will do our best to produce partial results during the discussion period, but we cannot guarantee that a complete implementation will be ready before the rebuttal deadline. If progress permits, we will include the additional results and analysis in the final version.

---

> > > > ### Author Response · Authors · 2025-11-29
> > > > **Response on the Feasibility of Incorporating SPF as a Baseline**
> > > >
> > > > “See, Point, Fly” (SPF) is a training-free and geometry-aware navigation framework, consisting of three main steps:
> > > >
> > > > 1. A vision-language model (VLM) predicts the target location in the current image, including its 2D pixel coordinates and estimated depth;
> > > >
> > > > 2. The predicted 2D point is then transformed into a 3D displacement vector via 2D-to-3D unprojection, resulting in an ActionPoint relative to the UAV’s current pose;
> > > >
> > > > 3. Based on the ActionPoint, low-level control commands are generated to drive the UAV through the environment.
> > > >
> > > > In contrast, CityNav is a navigation task constrained to a 2D plane (x, y). Adapting the SPF framework to CityNav would require the following modifications:
> > > >
> > > > - Only the 2D point prediction is retained from Step 1, and depth estimation is removed;
> > > >
> > > > - The vertical component (z-axis) of the ActionPoint is fixed to zero, reducing it to a 2D displacement;
> > > >
> > > > - The motion planning module only needs to handle simple 2D vector mapping for control.
> > > >
> > > > While this adaptation is technically feasible, it removes the core innovation of SPF—namely, the use of VLMs for 3D spatial reasoning and geometric 2D-to-3D unprojection. The adapted version no longer reflects SPF’s strengths in 3D spatial understanding and dynamic closed-loop UAV control in complex environments.
> > > >
> > > > Therefore, we believe that directly migrating SPF to CityNav as a baseline offers limited value and fails to capture the original design contributions of the SPF framework.

---

### Official Review · Reviewer_VBWr · 2025-10-28

**Soundness:** 3
**Presentation:** 3
**Contribution:** 2
**Rating:** 6
**Confidence:** 3

**Summary:**

This paper presents DAD-SFT, a novel framework for lightweight UAV Vision-Language Navigation (VLN) that integrates Dual Attention Distillation into Supervised Fine-Tuning. The authors propose CAD module to aligns the student’s semantic focus and CAA module to boosting discriminative ability. Experiments on CityNav show that DAD-SFT outperforms traditional and VLM-based baselines while maintaining low computational cost suitable for edge deployment.

**Strengths:**

1. The combination of cross-Modal attention distillation and contrastive attention alignment module effectively guides the model's attention distribution, boosting model’s robustness and generalization in unseen environments.
2. This article explores a solution to effectively deploy the professional capabilities of large models on edge devices, greatly optimizing memory and inference speed without sacrificing accuracy.

**Weaknesses:**

1. The success rate is not significantly improved compared to the baseline in prior work and the CityNav paper (SR is 10.16 in val-seen), raising doubts about whether VLM is a suitable baseline for the CityNav task.
2. The paper emphasizes the advantages of its lightweight model in deployment efficiency and provides an overall inference latency (6.82 seconds/step). However, this is an aggregate latency and does not provides a breakdown analysis on specific modules (perception acquisition , target inference or action planning.
3. The representation of the Semantic Map and the usage in the model are not provided.

**Questions:**

Whether the experiment is deployed on a real UAV platform?

---

> ### Author Response · Authors · 2025-11-20
> **Q1. Limited improvement in success rate; concerns on whether VLM is a suitable baseline for CityNav**
>
> Regarding the concern on Success Rate (SR), we would like to clarify that detailed comparisons are already provided in Table 1 (Section 5) of the paper. Our method achieves an SR of 12.73% on the Validation Seen split, which significantly outperforms the mainstream baseline MGP (8.69%). Compared with LLM methods, our model also achieves consistent improvements across multiple evaluation splits. For instance, on Test Unseen, our model achieves 12.96%, surpassing the 11.98% of the 32B VLM baseline, while also achieving competitive or better performance in terms of NE, OSR, and SPL.
>
> In addition, our work primarily focuses on balancing deployment efficiency and task performance under resource-constrained conditions. We adopt a compact 3B model, which maintains low inference latency and memory usage, while still achieving performance comparable to—or even exceeding—that of the much larger 32B model. Therefore, although the SR improvement may appear moderate in absolute terms, the achieved performance is practically valuable and demonstrates the effectiveness of our approach under lightweight deployment settings.

---

> ### Author Response · Authors · 2025-11-20
> **Q2. Lack of detailed breakdown of inference latency across system modules**
>
> We thank the reviewer for pointing out the missing breakdown in inference latency analysis.
>
> In fact, the inference time reported in Table 4 (Appendix I) corresponds to the Goal Inference module, where the model predicts the target position based on multimodal inputs.
>
> To provide a more comprehensive view of the system’s efficiency, we will include a detailed breakdown of latency across different modules in the revised version. Based on tests conducted on an RTX 4090 GPU, the latency per step is as follows:
>
> - Perception Acquisition: 1.53 s/step
> - Goal Inference: 6.52 s/step
> - Action Planning: 4.23 × 10⁻³ s/step
> - Environment Interaction: 7.41 × 10⁻³ s/step
>
> As shown above, the majority of the overall latency lies in the Goal Inference module, which is expected due to the computational complexity of semantic reasoning and vision-language alignment performed by the VLM.
>
> The Perception Acquisition module also contributes a relatively high cost, mainly because it needs to collect and preprocess multimodal inputs (e.g., RGB images, the semantic map) in real-time before feeding them into the model. In contrast, the Action Planning and Environment Interaction modules are lightweight and deterministic, incurring negligible latency.
>
> We will incorporate this analysis and annotate the time distribution across modules in the revised paper to better demonstrate the efficiency and deployability of our system in real-world scenarios.

---

> ### Author Response · Authors · 2025-11-20
> **Q3. Lack of clear explanation on the representation and usage of the semantic map**
>
> We appreciate the reviewer’s attention to the semantic map.
>
> In Section 3.3.1 of the paper, we have briefly described the structure of the semantic map, but we acknowledge that the current presentation may not be sufficiently clear. We will revise the text to provide a more comprehensive explanation.
>
> The semantic map we use is a 2D image that integrates both spatial and semantic information. It includes the following components:
>
> - The UAV’s current position and orientation (indicated by an arrow)
> - The first-person camera view (highlighted by a yellow bounding box)
> - Key landmarks (highlighted using red masks)
>
> This semantic map serves as a visual input and is used alongside natural language instructions as input to the model. In the prompt template (Appendix C), we also explicitly guide the model to reason over the semantic map and the textual goal jointly.
>
> We will enhance the explanation in the revised version, including clearer visualizations and descriptions to help readers better understand the map’s role and how it is processed by the model.

---

> ### Author Response · Authors · 2025-11-20
> **Q4. Whether the experiment is deployed on a real UAV platform?**
>
> Our experiments are conducted in the CityNav simulation platform, which offers realistic city-level street layouts, rich semantic landmarks, and natural language navigation instructions. This benchmark is widely adopted for UAV VLN and enables systematic evaluation of generalization in dynamic environments.
>
> Although most experiments are based on simulation, our method is designed with real-world deployment feasibility in mind. For instance, our lightweight 3B model runs on a single RTX 4090 GPU with 13.5GB memory usage and 6.82 s/step inference latency, significantly more efficient than larger-scale models such as Qwen2.5-VL-32B (70GB, 53.42 s/step) or GPT-4o (9.73 s/step, cloud-based), indicating strong deployment potential.
>
> In addition, we have conducted preliminary real-world deployment experiments on a Tello UAV (TLW004). This platform is equipped with a RGB camera, Vision Positioning System, IMU, barometer, infrared sensor, and supports motion control via APIs. Initial experiments show that the system is capable of successfully completing multiple instruction-driven navigation tasks, validating the feasibility and practicality of our method in real UAV platforms.

---

> > ### Comment · Reviewer_VBWr · 2025-11-24
> >
> > Thank the authors for the response. My major concerns have been addressed.

---

> > > ### Author Response · Authors · 2025-12-01
> > >
> > > Thank you for your positive feedback. We are glad to hear that the major concerns have been addressed. We will incorporate the discussed clarifications and improvements into the final version of the paper. We sincerely appreciate your constructive suggestions throughout the review process.

---

### Official Review · Reviewer_EEFV · 2025-10-28

**Soundness:** 2
**Presentation:** 2
**Contribution:** 2
**Rating:** 2
**Confidence:** 4

**Summary:**

The paper addresses the challenge of deploying large, computationally expensive Vision-Language Models (VLMs) for Unmanned Aerial Vehicle (UAV) navigation on resource-constrained devices. Lightweight models, while efficient, typically suffer from poor performance and generalization. To solve this, the authors propose a Dual Attention Distillation into Supervised Fine-Tuning (DAD-SFT) framework. This method uses Cross-Modal Attention Distillation (CAD) to train a lightweight "student" model to mimic the semantic focus patterns of a powerful "teacher" model. Simultaneously, it employs Contrastive Attention Alignment (CAA), which uses diverse negative samples to enhance the student model's discriminative ability. When evaluated on the CityNav benchmark, the proposed DAD-SFT framework reportedly outperforms baseline methods in navigation accuracy.

**Strengths:**

1. The paper tackles the critical, real-world trade-off between model performance and computational efficiency. Finding a way to run powerful VLMs on "edge devices" like UAVs is a significant and practical challenge for the field.
2. The core contribution, DAD-SFT, is a novel framework that intelligently combines two distinct techniques: knowledge distillation (via CAD) and contrastive learning (via CAA).
3. The method achieves impressive results on the CityNav benchmark, consistently outperforming a wide range of baselines, including traditional methods (Seq2Seq, CMA, MGP) and other large VLMs (LLaMA, GPT-4o).
4. The authors provide an ablation study in Table 1 that validates their design. By comparing "Naive SFT" against "CAD-only," "CAA-only," and the full "DAD-SFT," they successfully demonstrate that both components are complementary and contribute to the final model's superior performance.

**Weaknesses:**

1. The proposed model is not trained on the first-person view (FPV) RGB-D observations that an actual UAV would perceive. Instead, the model is fed a "Semantic Map", which is a top-down view 2D image of the whole map. This departs from a strict FPV-only setting and can leak strong global priors about where the target is, unlike real FPV drones that only see local observations.
2. The paper's core experimental setup is fundamentally mismatched with the CityNav baselines it compares against, which, in my view, invalidates the primary performance claims. The final model reports large gains over these baselines while adding a global RGB map view input. These comparisons are therefore not strictly fair to the FPV‑only baselines.
3. All experiments stay within a single simulator/dataset. The experiments don't test DAD-SFT framework transfer to other UAV VLN settings to test its generalization.
4. The authors repeatedly claim that the student model surpasses the teacher (Qwen2.5-VL-32B) on SR. However, this comparison is potentially misleading, as the teacher serves mainly as a distillation source rather than a comparative baseline fine-tuned using the same SFT/CoT regimen. This weakens the conclusion that the student is intrinsically stronger rather than advantaged by additional supervision tailored to the task.
5. While there is a first-vs-last-layer attention distillation study, the work lacks the ablation of sensitivity to $λ_{attn}$, $λ_{contrast}$ and per-negative-type contribution in CAA, which are central to the claimed gains.

**Questions:**

Please see the weaknesses

---

> ### Author Response · Authors · 2025-11-20
> **Q1. The use of Semantic Map instead of pure FPV RGB-D input**
>
> We thank the reviewer for pointing out this issue.
>
> We would like to clarify that CityNav is not a strictly first-person view (FPV)-only dataset. Its task design explicitly supports the use of geo-aligned semantic maps as one of the navigation inputs. The semantic map we use is a structured map representation that integrates UAV’s current position, orientation, field-of-view, and geo-referenced landmarks. Such inputs can be realistically constructed using multi-sensor fusion or map APIs, and are feasible for real-world deployment.
>
> We will further clarify the definition of the semantic map and the model's input assumptions in the revised version to avoid potential misunderstandings.

---

> ### Author Response · Authors · 2025-11-20
> **Q2. The concern of inconsistency with CityNav baselines**
>
> We understand the reviewer’s concern regarding the fairness of comparisons.
>
> We would like to emphasize that all baseline models we compare against were trained and evaluated under the same input configuration, in which the semantic map (RGB map view) was included as one of the inputs.
>
> In fact, the original CityNav paper (in Appendix C, link: https://openreview.net/pdf?id=LjvIJFCa5J) explicitly states that models such as Seq2Seq, CMA, and MGP utilize map-view inputs. For VLM baselines, we used a consistent prompt format and multimodal input, which also includes the semantic map.
>
> Therefore, our experimental setting is fully aligned with existing baselines, and no unfair advantage in input modality exists.
>
> We have already clarified this point in Section 4.2 (Baselines), so that reviewers and readers can clearly understand the consistency of our evaluation setup.

---

> ### Author Response · Authors · 2025-11-20
> **Q3. The concern of lacking generalization evaluation**
>
> We appreciate the reviewer’s concern regarding the generalization capability of our method.
>
> The CityNav dataset already provides three well-established splits — Validation Seen, Validation Unseen, and Test Unseen — covering both familiar and entirely novel scenes. Our model consistently outperforms baselines across all these splits, especially under Test Unseen, demonstrating its strong generalization ability.
>
> Furthermore, we fully agree that cross-dataset evaluation is crucial for comprehensively assessing generalization. We plan to extend our experiments in the future by adapting the model to other UAV VLN benchmarks, in order to further validate the general applicability of our method.

---

> ### Author Response · Authors · 2025-11-20
> **Q4. The clarity of the claim that the student model outperforms the teacher**
>
> We thank the reviewer for pointing out this issue.
>
> We acknowledge that the current claim that the student model “outperforms the teacher” may be misleading, as it does not clearly distinguish the difference in training conditions between the two.
>
> We will revise the statement to:
> The lightweight student model trained via the DAD-SFT framework achieves better performance and inference efficiency than the non-fine-tuned teacher model under the task setting of this work.

---

> ### Author Response · Authors · 2025-11-20
> **Q5. The sensitivity analysis of $\lambda_{\text{attn}}$, $\lambda_{\text{contrast}}$, and ablation on negative samples in CAA**
>
> We fully agree with the reviewer’s suggestion to conduct more detailed analysis on the hyperparameters and negative sample design. We have conducted additional experiments and will include the detailed results in the revised appendix. The main findings are summarized below:
>
> (1) Grid search sensitivity analysis on $\lambda_{\text{attn}}$, $\lambda_{\text{contrast}}$
>
> We perform a sensitivity analysis of the loss weights $\lambda_{\text{attn}}$ and $\lambda_{\text{contrast}}$ on the Validation Unseen split.
>
> The results (SR, %) are shown below:
>
> | $\lambda_{\text{attn}}$ / $\lambda_{\text{contrast}}$ | 0.05 | 0.1  | 0.2 | 0.5  |
> |-------------------------------|------|------|-------------|------|
> | 10                            | 7.85 | 8.12 | 8.46        | 7.9  |
> | 30                            | 9.02 | 9.56 | 10.18       | 9.61 |
> | 50                            | 10.12| 10.26| **10.43**    | 10.01|
> | 80                            | 9.76 | 10.07| 10.22       | 9.88 |
>
>
> The results show that the configuration ($\lambda_{\text{attn}}$ = 50, $\lambda_{\text{contrast}}$ = 0.2) achieves the best performance.
>
> The model is more sensitive to $\lambda_{\text{attn}}$: when $\lambda_{\text{attn}}$ is too small (e.g., 10), the model underperforms, whereas a moderate range (50–80) yields more stable and superior results.
>
> This suggests that attention distillation loss is more effective when applied with sufficient strength.
> In contrast, the model is relatively robust to $\lambda_{\text{contrast}}$ within a reasonable range, with particularly strong stability observed between 0.1 and 0.2.
>
> (2) Contribution ablation of different negative sample types
>
> To understand the contribution of different negative sample types in CAA, we conduct ablation experiments under the CAA-only setting (i.e., without CAD).
>
> The following results are from the Test Unseen split:
>
> Table: Effect of different negative sample types in CAA (SR %) on Test Unseen
> | Configuration                                 | SR (%) |
> |----------------------------------------------|--------|
> | Naive SFT                                     | 5.79   |
> | **CAA-only (full negative types)**            | **9.79** |
> | CAA-only w/o Random Attention Negatives       | 9.12   |
> | CAA-only w/o Perturbed Attention Negatives    | 9.26   |
> | CAA-only w/o Adversarial Attention Negatives  | 8.87   |
> | CAA-only w/o Cross-instance Attention Negatives | 9.03 |
>
> The results indicate that all negative sample types positively contribute to performance. Notably, Adversarial Attention and Cross-instance Attention negatives are most impactful in improving the model’s discriminability and overall performance. Removing any of them leads to a performance drop, validating the necessity of diverse negative designs in CAA.
>
> We have incorporated the full results and discussion of these experiments into Section 5.2 (Ablation Study) and Appendix K (Additional Ablation Studies) to improve the transparency and scientific rigor of our method.

---

> ### Comment · Reviewer_EEFV · 2025-11-23
>
> Thank you for your responses and clarifications.
>
> However, after revisiting the CityNav paper and examining your code, there appears to be a fundamental misunderstanding regarding the definition of "maps" and the specific inputs used by the baselines.
>
> In the CityNav setup (https://openreview.net/pdf?id=LjvIJFCa5J) there are two kinds of visual inputs that represent different spatial scopes, as illustrated in Figure 11:
>
> 1. **Navigation map.** This is 5 channel image (‘view area map’, 'landmark map', ...) each one represents a certain mask in the whole navigation environment. Therefore, the agent knows the relative locations of the landmarks mentioned in the prompt but cannot access RGB or depth information outside its field of view. These channels can be found at the bottom right part of Figure 11 in CityNav paper.
> 2. **Current RGB-D observation.** It is the RGB and depth view of the agent's view area. This observation represents only the small part of the whole environment that the drone can see. These observation can be found at the upper left part of Figure 11 in CityNav paper.
>
> I disagree with the statement that the original CityNav paper states that Seq2Seq and CMA utilize map‑view inputs. In appendix C.1 and C.2 they specifically mention that these models process RGB and Depth observations, and description. They do not access a global map. That's why they did a separate ablation study with MapSeq2Seq by providing navigation map in addition to RGB-D observation.
>
> In your paper, the reported numbers for Seq2Seq and CMA appear to directly match with the CityNav paper numbers, where they were trained without navigation maps. But even the MapSeq2Seq and MGP won't be a fair comparison with your approach as they don't have access to the RGB view data outside of their view area.
>
> Regarding the “semantic map” you used as input. As you mentioned it integrates UAV’s current position, orientation, field-of-view, and geo-referenced landmarks. However it also contains the full bird’s‑eye view of the surrounding city block, not a cropped local patch which I think isn’t explicitly mentioned. This can be confirmed by your provided code, where you process whole block images like map_0_20250508154410520330.jpg in your dataset class for training.
>
> This confirms that the agent receives the full environmental context at step 0, not just navigation map used in CityNav. Which effectively turning the problem from navigation into a more map-reading task, which diverges from the standard FPV VLN framework established by the benchmark you are citing.

---

> > ### Author Response · Authors · 2025-12-01
> >
> > Thank you very much for the reviewer’s detailed and constructive feedback.
> >
> > 1. On the baseline settings
> >
> >  We appreciate the reviewer for pointing out the issues in the Seq2Seq and CMA baseline results.
> >  After careful re-checking, we indeed found that compared with Seq2Seq and CMA, the VLM-based methods utilized richer semantic map information, which could give them an advantage in comparison.
> >  To improve the fairness and completeness of the evaluation, we have updated the baselines in the revised version by adopting Seq2Seq with map and CMA with map as the latest baseline results. These results are taken directly from the newest CityNav paper and kept fully consistent (link: https://openaccess.thecvf.com/content/ICCV2025/papers/Lee_CityNav_A_Large-Scale_Dataset_for_Real-World_Aerial_Navigation_ICCV_2025_paper.pdf).
> >  Overall, these additions do not change the main conclusions of our paper. The updated results can be found in Table 1 of the revised paper.
> >
> > 2. On the comparability of baselines
> >
> >  We believe that Seq2Seq with map, CMA with map, and MGP are comparable to our method in terms of input settings, as they all incorporate semantic map information beyond the First-Person View (FPV). And CityNav itself is not strictly an FPV-only setting.
> >  Meanwhile, our baselines also include several VLM models whose input configurations are fully aligned with ours, ensuring that the effectiveness of DAD-SFT is fairly validated among methods of the same category.
> >
> > 3. On the practical feasibility of the task setup
> >
> >  Our task setting is practically feasible. All the inputs required by our model—including the full semantic map—can be obtained through multi-sensor fusion or map APIs in real UAV systems, making the proposed setup applicable for real-world deployment.
> >
> > Once again, we sincerely thank the reviewer for the valuable comments, which have helped us significantly improve the quality and clarity of our paper.

---

### Author Response · Authors · 2025-12-03
**Meta-Review Response Summary for DAD-SFT**

Dear Area Chair,

We sincerely thank you and all reviewers for the thoughtful evaluations and constructive feedback on our work. Throughout the rebuttal process, we have carefully addressed each reviewer’s concerns and substantially revised and clarified the manuscript. Below is a consolidated summary of the core contributions, reviewers’ comments, and our corresponding responses and improvements.

## 1. Summary of Core Contributions

In this work, we propose DAD-SFT (Dual Attention Distillation with Supervised Fine-Tuning), a framework designed to efficiently transfer the capabilities of large vision-language models (VLMs) to lightweight UAV navigation models, enabling deployment on resource‑constrained platforms. Specifically:
- We introduce Cross-modal Attention Distillation (CAD), which guides the student to focus on the teacher’s semantic attention patterns, enhancing cross-modal perception.
- We propose Contrastive Attention Alignment (CAA), which incorporates contrastive learning with diverse negative samples to boost discriminative ability and generalization.
- Extensive experiments on the CityNav benchmark demonstrate that DAD-SFT significantly outperforms mainstream baselines while maintaining high efficiency, robustness, and deployability.

## 2. Reviewer Evaluations and Consensus
- **Clear problem importance and practical relevance**: All four reviewers (EEFV, VBWr, 3tX4, wPUD) recognize that our work addresses a real and pressing challenge: deploying powerful VLMs on edge devices. They highlight the practical value and relevance of our solution.

- **Novel and effective methodological design**: Reviewers 3tX4 and wPUD note that attention-level distillation is intuitive and well‑suited for navigation tasks. Reviewer EEFV affirms the complementary and effective integration of CAD and CAA.

- **Comprehensive experiments and ablation studies**: Reviewers EEFV, 3tX4, and wPUD commend the completeness of our experiments, including CAD/CAA ablations and layer-wise attention distillation, which validate the robustness and contribution of the method.

- **Strong efficiency and deployment advantages**: Reviewers VBWr, 3tX4, and wPUD appreciate that our 3B student model achieves competitive or superior performance to a 32B teacher while running efficiently on a single RTX 4090 (6.82s/step), demonstrating practical deployability.

## 3. Key Reviewer Concerns and Our Responses
 **Additional Ablation Studies & Parameter Sensitivity Analysis**
- We expanded ablation studies to include:
  - A two-dimensional grid search over λ_attn and λ_contrast;
  - A contribution breakdown of all negative sample types (random, perturbed, adversarial, cross‑instance).
- These analyses provide clear evidence of the complementary effects and stability of CAD and CAA.

 **Inference Latency Decomposition**
- Following reviewer feedback, we added a detailed module-wise latency breakdown.
- This analysis identifies the main performance bottleneck and offers insights for future optimization, improving the completeness of our deployment evaluation.

 **Statistical Robustness via Multiple Random Seeds**
- To address concerns about whether the student outperforming the teacher is statistically robust, we conducted multi‑seed training and evaluation.
- The success rate (SR) shows minimal variation across seeds, and other metrics (NE, OSR, SPL) remain stable.
- This confirms that the gains from DAD-SFT are statistically significant and reproducible.

 **Fairness of Input Settings and Baseline Comparisons**
- Reviewer EEFV raised concerns about the use of a semantic map.
- We clarified that CityNav officially supports the semantic map input and updated baselines accordingly to “Seq2Seq with map” and “CMA with map.”
- We further clarified the input settings in the revised manuscript.

 **Transparency and Scale of CoT Data Construction**
- We added detailed data statistics (Appendix J) covering instruction length, reasoning length, spatial coverage, and semantic alignment.
- We also clarified filtering and answer‑replacement procedures to ensure data correctness.

 **Missing Comparisons with Certain Methods (FitNet, Vi-LAD, SPF, OpenFly)**
- We implemented FitNet for UAV‑VLN and reported its preliminary performance, which is slightly better than CAD-only but below CAA-only and DAD‑SFT.
- For Vi-LAD, SPF, OpenFly, we explained architectural incompatibilities and migration challenges under CityNav, documenting them as future directions.

## 4. Revisions and Additions to the Manuscript
In response to reviewer feedback, we have incorporated several key revisions into the paper, including baseline adjustments, extended ablation analysis, and clearer CoT data descriptions, among others.

## 5. Conclusion
We appreciate the Area Chair and reviewers for their time and valuable feedback. The rebuttal process has helped us clarify key aspects and improve the overall quality of our paper.

---

### Meta-Review · Area_Chair_ajw3 · 2026-01-10

**Summary:**

This paper, proposing a specialized VLM distillation method for UAV VLN, received mixed reviews and a range of important concerns. Reviewer EEFV had a number of important concerns including the fact that there was a mismatch (admitted later by the authors) in the setting where the VLM-based methods used a global map representation. While the authors included map-based versions in the tables now, there are still concerns as the reviewer responded stating that there might still be a mismatch in definitions. This is a crucial aspect that should be correct on original submission, and therefore suggests resubmission to a later venue in order to very clearly lay out what representations/inputs are used by prior methods and variants, and what the method proposed uses. Other concerns included lack of generalized evaluation on other benchmarks to see if the results are generalizable (not even cross-dataset but just evaluation on other benchmarks) which is shared by Reviewers 3tX4 and wPUD, and sensitivity to hyper-parameters. Reviewer wPUD had a number of concerns including comparison with more state-of-art VLA architectures.

Reviewer VBWr had a range of concerns as well (though positive score of 6), and one that I strongly share is that the paper is fundamentally about knowledge distillation (targeted towards a very specific domain of UAV VLN) yet does not do a comprehensive comparison to the large body of both non-attention and attention-based distillation methods. Without this type of analysis, it is difficult to understand the contributions of the proposed method, whether/if they take advantage of unique problem characteristics in UAV VLN, and indeed (because of the single-benchmark focus) whether the results generalize to even other VLN domains.

**Reviewer Concerns:**

Some of the reviewer concerns, such as sensitivity analysis of hyper-parameters, variance of results across multiple runs, etc. were resolved with experiments. However, the major concerns noted above, especially full clarity and fair comparison in terms of how/what map representations are defined and used, demonstration of generality to other VLN benchmarks (mentioned by three reviewers), and importantly comparison to the body of knowledge distillation research, were not well addressed (with just one baseline method added). Since the method is fundamentally about knowledge distillation, with a claim to uniqueness for the UAV VLN domain, this is a crucial missing element.

**Reviewer Scores:**

For the more negative reviewers (e.g. EEFV and wPUD) my assessment is that they would be unlikely to change their scores given the discussion (for one of the reviewers) and nature of the rebuttal which did not address major concerns.

As a result, the current version cannot be accepted and requires significant clarification/correction (from the beginning, to allow a full review of other aspects) as well as addressing the above unaddressed concerns.

---

### Decision · Program_Chairs · 2026-01-26

Reject